# Tousled-like kinase 2 targets ASF1 histone chaperones through client mimicry

Bertrand Simon[1], Hua Jane Lou[1], Clotilde Huet-Calderwood[1], Guangda Shi[1], Titus J. Boggon [1,2], Benjamin E. Turk [1✉] & David A. Calderwood [1,3✉]

Tousled-like kinases (TLKs) are nuclear serine-threonine kinases essential for genome maintenance and proper cell division in animals and plants. A major function of TLKs is to phosphorylate the histone chaperone proteins ASF1a and ASF1b to facilitate DNA replication-coupled nucleosome assembly, but how TLKs selectively target these critical substrates is unknown. Here, we show that TLK2 selectivity towards ASF1 substrates is achieved in two ways. First, the TLK2 catalytic domain recognizes consensus phosphorylation site motifs in the ASF1 C-terminal tail. Second, a short sequence at the TLK2 N-terminus docks onto the ASF1a globular N-terminal domain in a manner that mimics its histone H3 client. Disrupting either catalytic or non-catalytic interactions through mutagenesis hampers ASF1 phosphorylation by TLK2 and cell growth. Our results suggest that the stringent selectivity of TLKs for ASF1 is enforced by an unusual interaction mode involving mutual recognition of a short sequence motifs by both kinase and substrate.

[1] Department of Pharmacology, Yale School of Medicine, New Haven, CT, USA. [2] Department of Molecular Biophysics and Biochemistry, Yale University, New Haven, CT, USA. [3] Department of Cell Biology, Yale School of Medicine, New Haven, CT, USA. ✉email: ben.turk@yale.edu; david.calderwood@yale.edu

Dynamic assembly and disassembly of chromatin controls DNA replication, DNA repair, transcription, and cell division, making it essential for normal growth, development and differentiation. The fundamental repeating structural unit of chromatin is the nucleosome, a particle composed of ~147 bp of DNA wrapped around octamers of histone proteins[1]. Nucleosome assembly is a regulated multi-step process where deposition of a tetramer of histone 3 and 4 [(H3-H4)$_2$] on DNA is followed by the addition of two histone (H2A-H2B) dimers[1]. These processes are controlled by histone chaperones and nucleosome assembly factors that bind histones, mediate their interactions with DNA, and modulate histone post-translational modifications[2]. Histone chaperones are themselves often post-translationally regulated, providing mechanisms to control chromatin assembly and stability. Here, we investigate mechanisms regulating phosphorylation of the evolutionarily-conserved H3–H4 chaperone anti-silencing factor 1 (ASF1) by the protein kinase tousled-like kinase 2 (TLK2).

In humans there are two ASF1 paralogues (ASF1a and ASF1b) with high sequence conservation and overlapping functions in both replication-dependent and replication-independent chromatin assembly and disassembly[3–6]. ASF1 proteins are critical for S-phase progression and, in cooperation with other histone chaperones, play key roles in many processes including assembly of newly synthesized DNA into chromatin and histone recycling during transcription[7–10]. Loss of ASF1 disrupts global transcription, causes DNA replication defects, activates the DNA damage response and can trigger gross chromosomal rearrangements[11–15]. ASF1 consists of an N-terminal 155-residue immunoglobulin-like domain (ASF1-NT) followed by a presumably unstructured C-terminal tail[16–18]. Structural and functional studies have revealed how ASF1-NT engages histone H3–H4 dimers in a manner preventing heterotetramer formation[18–20]. ASF1 then transfers the H3–H4 dimer to either the chromatin assembly factor 1 (CAF-1) complex or the histone chaperone HIRA, which facilitate deposition of H3–H4 tetramers onto DNA respectively during replication-coupled and replication-independent nucleosome assembly[5,6,21,22]. Through the assembly of multi-protein complexes, ASF1 can also regulate post-translational modifications of histone proteins[21,23–25]. Thus, ASF1 function is central to regulated assembly, disassembly and modification of nucleosomes.

While the globular ASF1-NT domain directly binds to all of its known interaction partners, its more poorly conserved C-terminal tail also plays important roles. In yeast, the Asf1 C-terminal tail is highly enriched in acidic residues (>50% Asp/Glu), and its removal causes a >200-fold reduction in histone H3–H4 binding affinity[26]. The corresponding region of plant and animal ASF1 orthologs is substantially shorter and less acidic, yet is a hotspot for phosphorylation, allowing regulation of the net charge of the tail. Specifically, phosphorylation of the C-terminal tail of histone-free ASF1 by tousled-like kinases (TLKs) increases its association with the H3–H4 dimer and downstream chaperones, which has been suggested to promote the provision of histones for chromatin assembly[3].

Mammals have two TLKs (TLK1 and TLK2), nuclear serine-threonine kinases whose activity fluctuates through the cell division cycle, peaking in S phase[27]. Their best-characterized roles are to positively regulate ASF1 to meet demand for increased nucleosome production during DNA replication[28–38], but they are also reported to regulate kinases involved in mitotic progression[29,33,35], to mediate DNA damage-induced cell cycle checkpoints[37,38] and to influence transcriptional silencing[28,31,32,34]. TLK2 depletion causes replication fork collapse and consequent DNA damage[39]. TLKs have been implicated in human disease in multiple contexts. Heterozygous loss of function mutations in TLK2 are associated with a disorder characterized by mild neurodevelopmental delay,

behavioral abnormalities, and facial dysmorphia[40,41]. TLK2 was also identified through an RNA interference screen as a suppressor of reactivation of gamma-herpes viruses from latency[42]. Finally, TLKs are amplified in multiple types of human cancer, and, in some cases, this is associated with poor prognosis[3,43]. Silencing TLK2 expression in particular reduces tumor cell growth in vitro and in mice and sensitizes cells to DNA damaging agents and checkpoint inhibitors[39,44]. High expression of TLKs also appears to suppress inflammatory signaling that may promote immune evasion by tumor cells[45]. These observations suggest that inhibition of TLKs may offer therapeutic benefit in treating cancer[39,44,46].

ASF1a and ASF1b appear to be the major phosphorylation targets of TLKs, at least during S phase. How TLKs selectively target ASF1 among thousands of other nuclear proteins is not completely clear, but protein kinases achieve substrate specificity in various ways, including recognition of consensus phosphorylation site motifs by the catalytic domain and recruitment of substrates through "docking" interfaces separate from the catalytic cleft[47]. Histone deficiency enhances phosphorylation of ASF1 by TLKs, and an intact histone-binding pocket on ASF1a is required for ASF1/TLK association, leading to a model where TLKs dock onto ASF1 via the histone H3/H4-binding site[18–20,48]. Here, we elucidate both catalytic site and non-catalytic interactions important for TLK2 phosphorylation of ASF1. We used an arrayed positional scanning peptide library to define a TLK2 consensus sequence that we verified to be important for optimal phosphorylation of ASF1. We further used deletion mapping, mutagenesis and X-ray crystallography to establish that a conserved N-terminal TLK2 sequence binds to the ASF1-NT domain in a binding mode similar to that of histone H3. Our observations suggest that specific TLK-ASF1 interactions are mediated by short sequence motifs found on both the kinase and substrate.

## Results

**TLK2 phosphorylation site specificity promotes phosphorylation of the ASF1 C-terminal tail.** To understand how TLK2 selectively targets ASF1, we initially analyzed its phosphorylation site specificity using an arrayed positional scanning peptide library (PSPL)[49]. The PSPL consisted of 182 peptide mixtures in which each of nine positions surrounding a central phosphorylation site were systematically fixed as one of the twenty amino acids, as well as two peptide mixtures in which the phosphoacceptor residue was fixed as either Ser or Thr. For these experiments we used a truncated form of TLK2 that removes the N-terminal 178 residues (TLK2 ΔN178), which we could robustly express and purify in active form from mammalian cells. The PSPL was incubated with TLK2 and radiolabeled ATP, and the level of phosphorylation of each component was quantified (Fig. 1a, b and Supplementary Table 1). TLK2 displayed an unusual phosphorylation site motif driven by strong selection for acidic residues upstream of the phosphorylation site, most prominently at the −2 position. In addition, TLK2 selected aliphatic and amidic residues at the +1 position immediately downstream of the phosphorylation site. Residues selected by TLK2 were overrepresented in previously reported phosphorylation sites on ASF1a and ASF1b (Fig. 1c)[48,50]. To verify the importance of residues proximal to the phosphorylation site, we designed a synthetic consensus peptide substrate incorporating residues most highly selected in the peptide library screen and examined the impact of point substitutions at key positions on TLK2 kinase activity (Fig. 1d). As anticipated from the PSPL analysis, we found that substitution of the −2 Glu and +1 Asn residues caused a significant decrease in the rate of phosphorylation. We found however that there was no significant difference in phosphorylation rate when the −1 Glu residue was

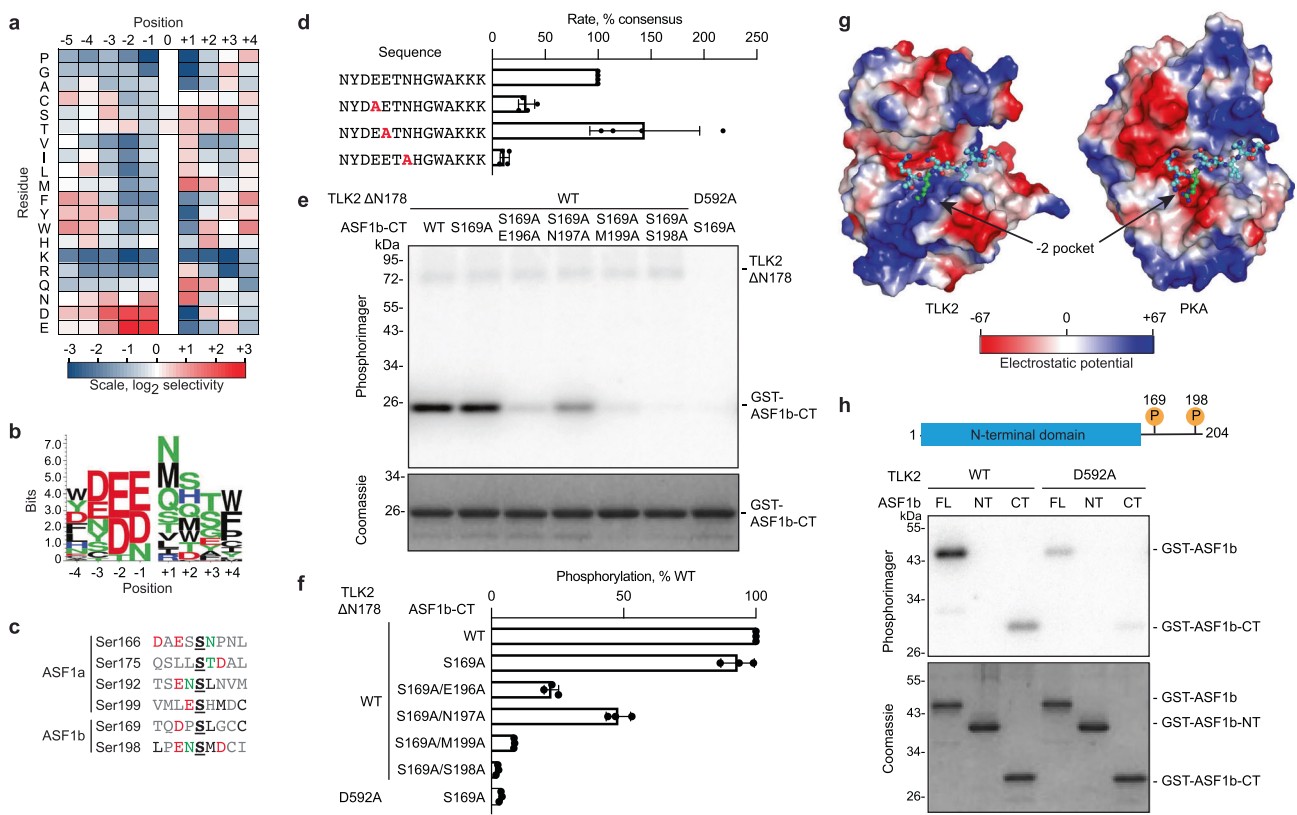

**Fig. 1 Optimal TLK2 phosphorylation of the C-terminal tails of ASF1 proteins is driven by both phosphorylation site specificity and the NT domain of ASF1. a** Heat map depicting amino acid selectivity at positions surrounding a Ser/Thr phosphorylation site determined from arrayed PSPL analysis of TLK2 ΔN178. The heat map shows normalized, log2 transformed PSPL spot intensities averaged across two separate experiments. Numerical values are provided in Supplementary Table 1. **b** Sequence logos showing positively selected residues were generated from the data shown in **a** using Seq2Logo[91]. **c** Sequences of previously reported phosphorylation sites on human ASF1a and ASF1b. **d** Relative rates of TLK2 ΔN178 phosphorylation of a synthetic consensus peptide substrate (top) and Ala substitution variants (bars show mean ± SD, $n = 4$ independent experiments). **e** Representative assay showing TLK2 ΔN178 phosphorylation of purified GST-tagged ASF1b-CT (residues 155–202) and ASF1b-CT mutants was assessed in kinase assays in vitro. A kinase inactive TLK2-D592A mutant was included as a control. **f** Quantified results from three independent replicates of the experiment shown in panel **e** (mean ± SD) (right). **g** Electrostatic surface potential of the TLK2 and PKA catalytic domains (PDB 5O0Y[51] and 1ATP[52]) with overlayed peptide fragment from the PKA-PKI complex (PDB 1ATP) generated using Pymol. **h** Phosphorylation of purified GST-tagged FL ASF1b (FL), the N-terminal domain (NT, residues 1–155) or the C-terminal tail (CT, residues 155–202) by purified FL TLK2 (WT or kinase inactive D592A mutant) was assessed using in vitro kinase assays. Results from one representative experiment of two are depicted, along with a cartoon depicting ASF1b domain organization and major sites of phosphorylation. Source data are provided as a Source Data file.

changed to Ala. This observation may reflect redundancy among the three acidic residues upstream of the phosphorylation in the peptide, suggesting that the importance of specific residues near the phosphorylation site likely depends on the overall sequence context. To determine whether residues selected by PSPL are important in the context of an authentic TLK2 substrate, we mutated residues flanking ASF1b Ser198, which conforms closely to the experimentally determined motif. These experiments were done in the context of a truncated form of ASF1b that included only the unstructured C-terminal region (ASF1b-CT, residues 155–202) expressed as a GST fusion protein. To avoid background phosphorylation at other sites, we mutated Ser169, the minor ASF1b phosphorylation site, to Ala. While this mutation alone had no apparent effect on TLK2 ΔN178 phosphorylation of ASF1b-CT, a S169A/S198A double mutant reduced phosphorylation approximately 50-fold, verifying Ser198 to be the major site of phosphorylation (Fig. 1E, F). A similar reduction in ASF1b-CT-S169A phosphorylation was observed when a kinase inactive form of TLK2 ΔN178 (D592A) was used, indicating that our WT preparations are not contaminated with a high level of other co-purifying ASF1b kinases. We found that mutation of the −2 Glu, −1 Asn, or +1 Met residue of ASF1b-CT reduced

phosphorylation by TLK2. Similar to our observations with the synthetic peptide substrate, the largest effect came from mutating the +1 residue. Overlaying the published structure of the TLK2 catalytic domain[51] with that of cAMP-dependent protein kinase (PKA) in complex with a peptide substrate[52] revealed an overall positive electrostatic surface potential in the region predicted to interact with residues upstream of the phosphorylation site (Fig. 1g). By contrast, PKA, which selects basic Arg residues at the −2 and −3 positions[53], had a net negative potential in the corresponding region. Multiple basic residues in helix αD (Lys557 and Lys560) and in the αF-αG loop (Arg678 and Lys679) of TLK2 contribute to the net positive charge in this region that likely promotes selectivity for acidic residues.

These observations suggest that efficient phosphorylation by TLK2 of ASF1 proteins is driven, at least partly, by recognition of favorable sequences surrounding their sites of phosphorylation. However, TLK1 has also been proposed to interact with the histone-binding site of ASF1[48], located in the ASF1 NT-domain. Consistent with this suggestion, we found that when using full-length (FL) TLK2 as the kinase, phosphorylation of ASF1b-CT was substantially lower than that of FL ASF1b (Fig. 1h). We note that in these experiments there was substantial background

activity in preparations of the FL TLK2 kinase inactive D592A mutant, presumably reflecting increased co-purification with the endogenous kinase due to oligomerization with the N-terminal coiled-coil regions[51]. This result suggests that optimal phosphorylation of ASF1 proteins requires a docking interaction occurring between the ASF1-NT and an uncharacterized region of TLK2.

**The non-catalytic N-terminal region of TLK2 interacts with ASF1a**. As a first step toward identifying the ASF1-binding site in TLK2, we evaluated the ability of FLAG-tagged ASF1a to co-purify with N-terminally GFP-tagged TLK2 (GFP-TLK2) in HEK 293T cells. Cells were transfected with wild-type (WT) or kinase inactive (D592A) GFP-TLK2, GFP-tagged proteins were pulled down using GFP-nanotrap coated beads[54] and co-purification of co-expressed FLAG-tagged ASF1a was determined by immunoblot. Both WT and kinase dead TLK2 pulled down ASF1, while GFP alone did not (Fig. 2a). ASF1 phosphorylation results in a shift in electrophoretic mobility[30,48] and TLK2 drives ASF1 towards the slower migrating phosphorylated form (Fig. 2a). As previously reported[30,51], the kinase inactive mutant of TLK2 consistently pulled down significantly more ASF1a than did WT TLK2 (Fig. 2b), and we therefore used this mutant for subsequent pull-down experiments. TLK2 is composed of a presumably unstructured N-terminal region followed by three putative coiled-coils and a C-terminal kinase domain[3,51] (Fig. 2c). Therefore, to map the ASF1a-binding site in TLK2 we used a series of C-terminal truncation mutants that eliminate the kinase domain (1–451), the kinase domain plus each coiled-coil (1–396, 1–296, 1–214) or that retain only the N-terminal half of the unstructured N-terminus (1–123) (Fig. 2c). As shown in Fig. 2d, and quantified in Fig. 2e, all five deletion constructs pulled down ASF1a at levels comparable to the FL TLK2 protein. The GFP tag alone did not pull-down detectable levels of ASF1a. These data suggest that the first 123 residues of TLK2 are sufficient for the interaction with ASF1a. We confirmed this result using a series of N-terminal TLK2 truncations. All constructs that remove the first 123 residues reduced ASF1 binding to control GFP levels (Fig. 2f, g). Thus, we conclude that the first half of the N-terminal region of TLK2 is both necessary and sufficient for its interaction with ASF1a.

**A short peptide from the TLK2 N-terminus directly binds the ASF1a-NT domain**. The co-precipitation assays from mammalian cells strongly implicate the first 123 amino acids of TLK2 in binding ASF1a. Multiple sequence alignments of TLK1 and TLK2 proteins across a broad range of animal species (Supplementary Fig. 1) showed strong conservation of the C-terminal coiled-coil motifs and the kinase domain. While the N-terminal region was less conserved overall, we noted a stretch of amino acids at the N-terminus (residues 2–22 in human TLK2) that was nearly identical among vertebrate TLK1 and TLK2 homologs. We therefore tested whether a GFP-TLK2 construct lacking this region (TLK2 24–772) bound ASF1a. While kinase-inactive full-length TLK2 pulled down endogenous ASF1a, deletion of the N-terminal 23 residues strongly inhibited binding to the background levels seen with the TLK2 123–772 construct or GFP alone (Fig. 3a, b). This strongly suggests that the ASF1 binding site lies in the N-terminal conserved 23 amino acids of TLK2. To test this idea further, we used differential scanning fluorimetry (DSF)[55] to determine the melting temperature of the ASF1a-NT domain (ASF1a residues 1–155) in the presence or absence of a chemically synthesized TLK2 peptide spanning residues 3–23. The addition of TLK2 peptide at a final concentration of 160 µM increased the melting temperature of purified recombinant ASF1a-NT by approximately 5 °C (Fig. 3c), consistent with peptide binding to and stabilizing of ASF1a-NT. We extended this result

using a fluorescence polarization (FP) assay to assess the interaction of the TLK2 peptide with ASF1a-NT. We measured the ability of unlabeled TLK2 peptide to displace fluorescently labeled TLK2 peptide from ASF1a-NT by following the resultant decrease in FP[56,57]. As shown in Fig. 3d, the FP signal follows a sigmoid curve with increasing concentrations of unlabeled peptide. The concentration of competing ligand necessary to displace 50% ($EC_{50}$) of TLK2 fluorescent peptide from ASF1a-NT was 2.2 µM (95% CI = 1.8–2.7 µM). Together our data indicate that the N-terminal TLK2 peptide specifically binds ASF1a-NT.

**Crystal structure of the ASF1a NT domain in complex with the TLK2 N-terminal peptide**. In order to determine how TLK2 engages ASF1a, we determined the crystal structure of the human ASF1a NT domain in the presence and absence of the N-terminal TLK2 peptide. Apo-ASF1a-NT crystallized in space group *C2* with seven molecules in the asymmetric unit, while the complex with the TLK2 peptide crystalized in the *P2₁* space group with eight complexes in the asymmetric unit. Both structures were determined by molecular replacement using the ASF1a molecule of the ASF1a·HIRA complex (PDB accession code 2I32 chain A)[6] as a search model and model bias was mitigated using Phenix Autobuild[58] (Table 1). The structure of apo-ASF1 NT domain was resolved to 2.1 Å and in complex with the TLK2 peptide to 2.71 Å. As expected from prior structures of human and yeast ASF1[19,20,59], ASF1a-NT adopted an immunoglobulin (Ig) domain-like fold, composed of a β sandwich topped by α helices (Fig. 4a). The ASF1a structure exhibited little variation between the chains in each structure, with an average root mean square deviation (RMSD) of 0.65 Å between 153–155 residues for the seven chains in the apo-ASF1a structure and RMSD of 0.39 Å over 151–154 residues among the eight ASF1a chains in the ASF1a·TLK2 peptide complex structure (Supplementary Fig. 2A, B). Notably, aside from re-orientation of side chains interacting with the TLK2 peptide, the ASF1a Ig-like domain did not display any significant structural changes between the apo and complex form (Fig. 4a, b), with an average RMSD of 0.59 Å over 151–154 residues between chain A of apo-ASF1a structure and the eight ASF1a chains of the ASF1a·TLK2 structure.

Prior structures of ASF1 in complex with histones and other co-chaperones[5,6,18,20,21,24,59–61], have revealed three main interaction interfaces: the H3-binding site around ASF1a residue Val94 that interfaces with two α-helices in the H3 dimer, a separate pocket that accommodates H4 strand βC, and a hydrophobic cleft on the opposite face from the H3 binding site that forms β-sheet interactions with the B-domain of HIRA and other co-chaperones[6,11,21] (Fig. 4c). Our ASF1a·TLK2 peptide complex structure data contained additional electron density absent from the apo-ASF1a structure, consistent with peptide binding. Surprisingly, additional density was observed at two of the previously characterized binding surfaces, the H3 binding site and at the B-domain binding site (Fig. 4b and Supplementary Fig. 3A–D). Phenix Autobuild completed those electron density pockets with the TLK2 peptide sequence, and both peptides were retained during refinement (Fig. 4b). As discussed in more detail below, these results pointed to the association of two TLK2 peptides per ASF1 domain in the crystal rather than a single peptide interacting at two sites.

In the ASF1a·TLK2 peptide complex, peptide binding at the H3-binding site is observed in all eight copies of ASF1a-TLK2 residues spanning Asp9 to Gly22 can be built into each copy, with residues His6 to Val23 visible in some cases. Binding buries on average 1270 Å$^2$ of both proteins surface area across the eight complexes. The bound TLK2 peptide adopts an α-helical secondary structure starting at residue Asp9 and ending at

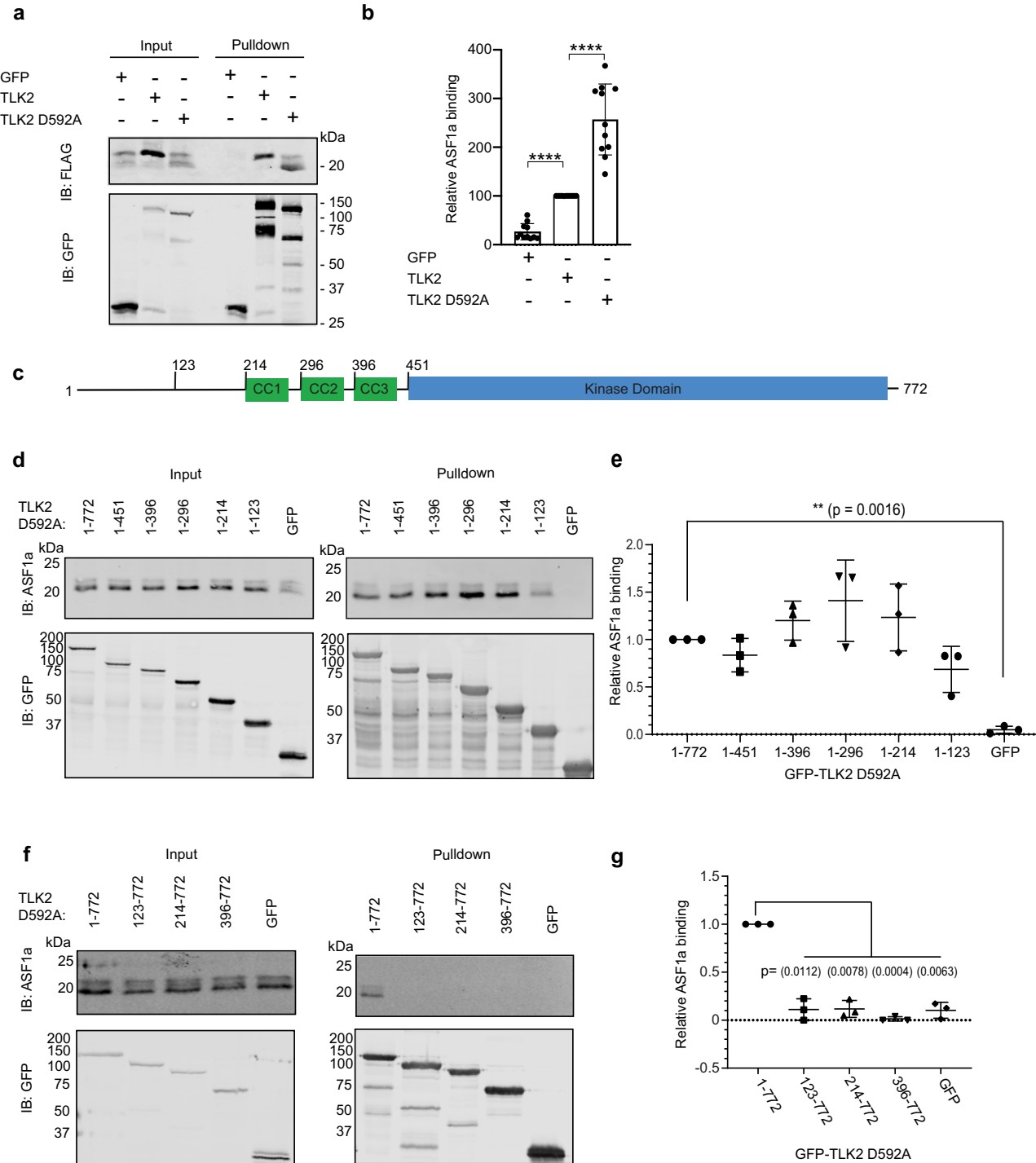

**Fig. 2 The non-catalytic N-terminal region of TLK2 interacts with ASF1a. a**, **b** GFP, GFP-TLK2, or kinase-inactive GFP-TLK2 (D592A) was co-expressed with FLAG-tagged ASF1a in HEK293T cells, cells were lysed and GFP proteins were precipitated with anti-GFP nanobodies coupled to Sepharose beads. Precipitated GFP proteins and co-precipitating FLAG-ASF1a were fractionated by SDS-PAGE and detected by immunoblotting. A representative experiment is shown in **a**. Relative ASF1a binding, normalized for FLAG-ASF1a input was quantified in 11 independent replicates, and mean ± SD is shown in **b**. Both unphosphorylated and phosphorylated forms of ASF1 were included in the analysis. ****$p$ values < 0.0001 in one-way ANOVA with Tukey's multiple comparison test. **c** Cartoon depicting TLK2 domain organization. Amino acid numbers are indicated. **d**–**g** GFP-TLK2 (D592A) and N-terminal or C-terminal TLK2 deletion constructs were expressed in HEK293T cells, GFP-nanobody pulldowns were performed as in **a** and co-precipitating endogenous ASF1a was detected by immunoblotting. Representative experiments are shown in **d** and **f**. and mean ± SD of three independent replicates are quantified in **e** and **g**. Significant difference from GFP-TLK2 D592A (1–772) calculated in one-way ANOVA with Dunnett's multiple comparison test—$p$ values shown on graph. Source data are provided as a Source Data file.

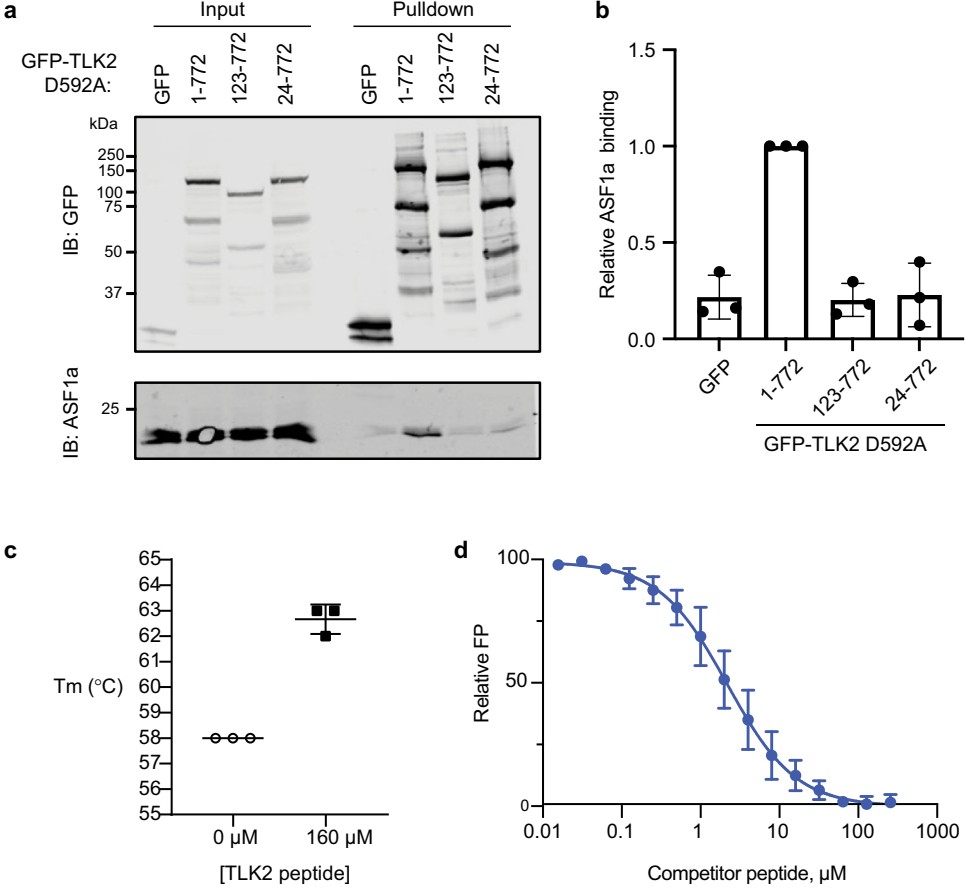

**Fig. 3 A short peptide from the TLK2 N-terminus directly binds the ASF1a NT domain. a, b** GFP, GFP-TLK2 (D592A) full-length (1–772) or N-terminal truncations (123–772 or 24–772) were expressed in HEK293T cells. Precipitated GFP proteins and co-precipitating endogenous ASF1a were fractionated by SDS-PAGE and detected by immunoblotting. A representative experiment is shown in **a**. Relative ASF1a binding normalized to full-length GFP-TLK2 (D592A) was quantified in three independent replicates and mean ± S.D is shown in **b**. **c** DSF was used to determine the melting temperature of purified ASF1a 1–155 in the presence or absence of 160 μM TLK2 peptide spanning residues 3–23. Mean ± SD of three independent replicates are shown. **d** FP assay to assess the interaction of the TLK2 peptide with purified recombinant ASF1a-NT. Increasing concentrations of unlabeled TLK2 peptide were titrated into a mixture of a fixed concentration of fluorescently labeled TLK2 peptide and purified recombinant ASF1a-NT, and FP was assessed. Results are shown as mean ± SD of independent experiments ($n = 3$ for competitor peptide concentrations 0–32 μM, $n = 2$ for 64–256 μM). Source data are provided as a Source Data file.

**Table 1 Data collection and refinement statistics (molecular replacement).**

|  | ASF1a TLK2 complex (7LO0) | Apo-ASF1a (7LNY) |
|---|---|---|
| Data collection |  |  |
| Space group | $P2_1$ | C2 |
| Cell dimensions |  |  |
| $a, b, c$ (Å) | 88.0, 136.7, 101.3 | 142.3, 119.0, 109.5 |
| $\alpha, \beta, \gamma$ (°) | 90.0, 103.2, 90.0 | 90.0, 103.5, 90.0 |
| Resolution (Å) | 98.63 − 2.71 (2.78 − 2.71)[a] | 106.5 − 2.10 (2.14 − 2.10)[a] |
| $R_{merge}$ | 5.7 (43.9) | 6.1 (69.8) |
| $I/\sigma I$ | 10.9 (0.85) | 12.8 (1.64) |
| Completeness (%) | 94.8 (89.0) | 98.4 (93.4) |
| Redundancy | 3.2 (3.0) | 4.4 (4.5) |
| Refinement |  |  |
| Resolution (Å) | 98.63 − 2.71 | 106.5 − 2.10 |
| No. reflections | 62,010 (3453) | 102,489 (6287) |
| $R_{work}/R_{free}$ | 21.3 (42.5)/22.6 (47.3) | 22.0 (35.6)/24.2 (31.5) |
| No. atoms |  |  |
| Protein | 22,296 | 16,951 |
| Water | 166 | 465 |
| B-factors |  |  |
| Protein | 63.2 | 56.5 |
| Water | 47.4 | 60.1 |
| R.m.s. deviations |  |  |
| Bond lengths (Å) | 0.0024 | 0.002 |
| Bond angles (°) | 0.515 | 0.518 |

A single crystal was used to determine each structure.
[a]Values in parentheses are for highest-resolution shell.

residue Thr21 (Fig. 4d). The overall mode of interaction is similar to how ASF1a engages helix α3 of histone H3 (Fig. 4e) as seen in ASF1a·H3/H4 structures[18,20,60]. The interaction at this interface is mediated by polar contacts between Arg12 and Arg19 of TLK2 (analogous to Lys122 and Arg129 in H3) and ASF1a residues Asp88 and Asp54, respectively. Additional polar contacts are made between the sidechains of ASF1a Arg108 and Thr147 and main chain groups of Arg19 and Phe20 near the C-terminus of the TLK peptide (or Arg129 and Arg131 of H3). Furthermore, Leu16 and Phe20 of TLK2 (like Leu126 and Ile130 in H3) make van der Waals interactions within the hydrophobic pockets centered on ASF1a Val94 and including Ala48, Leu96, and Tyr112 (Fig. 4d). In addition to these conserved interactions, TLK2 makes additional contacts not observed in ASF1a/ H3 structures, including an ion pair between Arg11 of TLK2 and Glu49 of ASF1a. In three chains, Leu8 of TLK2 is also visible, and it extends the network of hydrophobic interactions of TLK2 Leu16 and Phe20 to include Val92 of ASF1a (Supplementary Fig. 3E). With the exception of Arg11, residues at the TLK2- ASF1a binding interface are widely conserved in the N-terminal region of TLK orthologs across most animal and plant phyla as well as some protists, though the sequence is absent from several animal orders including flies and annelids. (Supplementary Fig. 4A). The TLK2 N-terminal sequence can be aligned with

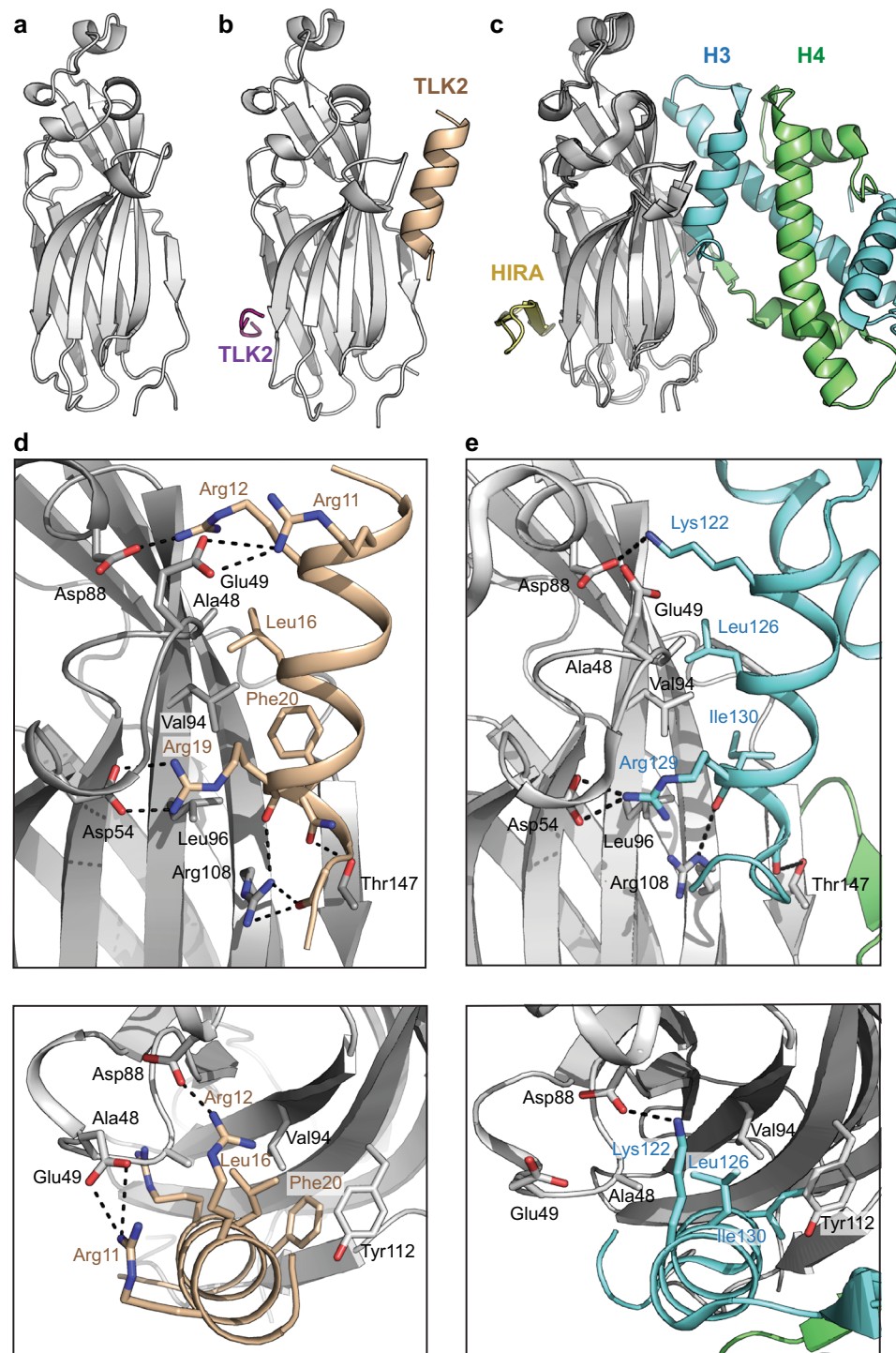

**Fig. 4 Structural basis of TLK2 binding to ASF1a.** Cartoon depictions of the structure of one molecule of the NT domain of human ASF1a (gray) (**a**) or of one molecule of the NT domain of human ASF1a bound to N-terminal peptides from TLK2 (ASF1a—gray; TLK2—beige and purple) (**b**). **c** Cartoon depicting known interaction surfaces on ASF1a and their interaction with partners based on PDB code 2I32[6] and 2IO5[20]. ASF1a—gray, histone H3—cyan, histone H4—green, HIRA—yellow. **d**, **e** Detail of TLK2 binding ASF1a at the H3-binding site (**d**) and comparison with the equivalent ASF1a-H3-H4 interaction (**e**), colored as in **b** and **c**.

the analogous ASF1-binding region from histone H3, and structural alignment (Supplementary Fig. 4B, C) revealed that conserved residues occupy analogous positions at the ASF1 interaction surface. Overall, our structural analysis suggests that TLK2 binds in a manner similar to how ASF1 engages histone H3, and hence that TLK2 would not bind to H3/H4-bound ASF1.

As noted earlier, the ASF1a·TLK2 complex structure also exhibited extra density at the B-domain binding site of ASF1a (Fig. 4b and Supplementary Fig. 3B–D). However, unlike the peptide modeled at the histone H3 binding site, here the visible region is comparatively short, involves some non-conserved residues, and it lacks hydrophobic and polar contacts essential for the HIRA–ASF1a interaction (Supplementary Fig. 4D, E).

The TLK2 peptide bound at this site is furthermore integrated in the crystal lattice interface of the asymmetric unit, suggesting that it may be an artifact (Supplementary Fig. 4F). Overall, the TLK2 N-terminal peptide binding at this interface does not appear homologous to HIRA, while its binding at the H3 binding site is strikingly similar to what is observed for H3 in terms of sequence, structure and side chain interactions.

**TLK2 binds at the H3-binding site of ASF1.** To test whether the crystallographically observed interaction sites are relevant in solution we generated structure-guided TLK2 and ASF1a mutants for use in binding assays. Specifically, the following mutants were introduced in the FL GFP-TLK2 (D592A) expression construct: L8R, R11A, R12A, and a triple mutant L16A/R19A/F20A (LRF/AAA). Each construct was co-transfected with FLAG-tagged ASF1a, GFP-TLK2 was pulled down with the GFP nanotrap coated beads, and ASF1a was detected by immunoblot as described earlier. While the mutation of Leu8 to arginine did not significantly alter binding, R11A, R12A, and the triple mutant LRF/AAA each prevented ASF1a from co-purifying with TLK2 (Fig. 5a, b). Mutation of individual residues within the triple mutant established that L16A or R19A mutations strongly inhibited binding while the F20A mutation had a slightly weaker effect (Supplementary Fig. 5). These results further confirm the importance of residues in the N-terminal region of TLK2 for binding ASF1a. However, they do not unequivocally resolve whether TLK2 binds at the H3-binding site or the B-domain binding site, as Arg12 and Leu16 are involved in both interfaces. To discriminate more stringently between the two sites, we introduced mutations into the ASF1a expression construct and performed co-immunoprecipitation experiments with co-expressed GFP-TLK2. Mutations at the ASF1a B-domain-binding site (D37A and D58A) had no significant impact on its ability to associate with GFP-TLK2 (Fig. 5c, d). In contrast, mutations in the H3-binding site (E49A, D88A, and V94R) strongly inhibited binding to GFP-TLK2 (Fig. 5c, d). Additional support for the importance of the H3-binding site interface on ASF1a in TLK2 binding comes from the observation that mutation of residues analogous to the H3 interacting residues (Arg12, Leu16, Arg19, and Phe20) impaired binding more strongly than mutation of the residue unique to only some animal TLK2 orthologs (Arg11). Finally, we compared the abilities of ASF1-binding histone H3.1 and HIRA B-domain peptides to compete with fluorescently labeled TLK2 peptide for binding to ASF1a-NT in fluorescence polarization assays. While the H3.1 peptide was an effective competitor (Fig. 5e), neither the B-domain peptide nor a TLK2 L16A/R19A/F20A triple mutant peptide could detectably displace the labeled TLK2 peptide. Taken together, our results clearly indicate that TLK2 binds at the H3-binding site and not the B-domain binding site.

**TLK2 binding at the H3-binding site promotes ASF1 phosphorylation and cell growth.** As noted above (Fig. 1h), optimal phosphorylation of ASF1b by TLK2 requires the ASF1b N-terminal region. We therefore investigated whether the crystallographically-defined TLK2–ASF1a binding interface serves to promote ASF1a phosphorylation. To do so, we examined the impact of mutating key residues in ASF1a on phosphorylation by TLK2 in vitro. Similar to ASF1b above, we found that FL TLK2 phosphorylated FL ASF1a to a much greater extent (~80-fold) than ASF1a-CT (Fig. 6a, b). As we observed in binding assays, mutation of the B-domain binding site in ASF1a (D37A) did not reduce its phosphorylation by TLK2. In contrast, mutation of the H3 binding interface (D88A or V94R) caused a significant decrease in the level of phosphorylation. In parallel,

we purified TLK2 with the LRF/AAA mutation that showed loss of binding in co-purification experiments. In comparison to WT TLK2, TLK2 (LRF/AAA) phosphorylated FL ASF1a only modestly (4-fold) better than ASF1a-CT. In addition, there was little impact (<2-fold) of mutating the ASF1a H3 binding interface on phosphorylation by TLK2 (LRF/AAA). Thus, the TLK2 N-terminal region, by binding the H3-binding site on ASF1, is necessary for optimal ASF1 phosphorylation in vitro.

To confirm the importance of the TLK2-ASF1 interaction on ASF1 phosphorylation in cells, we first co-expressed FLAG-tagged ASF1a with GFP-tagged WT or mutant TLK2 in HEK 293T cells. FLAG-ASF1a phosphorylation, assessed by mobility shift on SDS-PAGE, was significantly increased by expression of GFP-TLK2, but not by the kinase-inactive mutant D592A (Fig. 7a, b). Notably, co-expression with the ASF1-binding defective TLK2 mutant (LRF/AAA) provided a significantly reduced mobility shift compared to TLK2 WT, consistent with the important role of the docking interaction in promoting phosphorylation (Fig. 7a, b). To examine the importance of the docking interface to phosphorylation of endogenous ASF1a, we expressed various TLK2 alleles from an sgRNA-resistant cDNA in MCF7 cells in which we deleted TLK2 by CRISPR/Cas9 gene editing. As shown in Fig. 7c, loss of TLK2 significantly reduced phosphorylation of endogenous ASF1a, as assessed by electrophoretic mobility shift. Residual ASF1a phosphorylation in these cells likely reflects contribution from TLK1. While expression of WT TLK2 in these cells fully restored ASF1a phosphorylation, neither the kinase inactive D592A mutant nor the ASF1-binding defective LRF/AAA mutant rescued ASF1a phosphorylation (Fig. 7c, d). Furthermore, consistent with a prior report[44], we found that loss of TLK2 strongly suppressed clonogenic growth of MCF7 cells (Fig. 7e). In keeping with their impact on ASF1a phosphorylation, expression of WT TLK2, but not kinase inactive or ASF1-binding defective TLK2, restored colony growth (Fig. 7e, f).

Together our results establish that the TLK2 N-terminal region is necessary for interaction with and phosphorylation of ASF1a, acting as a docking motif for the kinase domain to facilitate phosphorylation of the C-terminal ASF1 tail. The importance of this interaction for phosphorylation is evident in vitro and in cells, where it is required for maximal phosphorylation of ASF1 and optimal cell growth.

## Discussion

Cell division, gene transcription, and genome maintenance rely on dynamic assembly and disassembly of nucleosomes. By binding histone H3–H4 dimers, the histone chaperone ASF1 facilitates nucleosome assembly and turnover and modulates histone post-translational modifications[10,21]. ASF1 activity is in turn regulated through phosphorylation by the nuclear kinases TLK1 and TLK2[48]. Here we have investigated the molecular basis for TLK2 specificity for ASF1.

The TLK1/2 catalytic domains are divergent from other kinases at the primary sequence level, with the human orthologs having only ~30% sequence identity to their closest relatives. Accordingly, TLKs have been classified as falling outside of the major eukaryotic kinase groups (e.g., AGC, CAMK, CMGC etc.)[62]. Consistent with their low similarity to other kinases, we found TLK2 to have a distinct phosphorylation site recognition motif, including selectivity for acidic residues at several positions upstream of the phosphorylation site and for amidic and aliphatic residues at the +1 position. This phosphorylation site motif is broadly similar to that previously reported for TLK1[63], consistent with their redundancy in phosphorylation of ASF1 and possibly other substrates[64]. Among previously characterized kinases, TLK1/2 are most similar to the polo-like kinases (PLKs), with

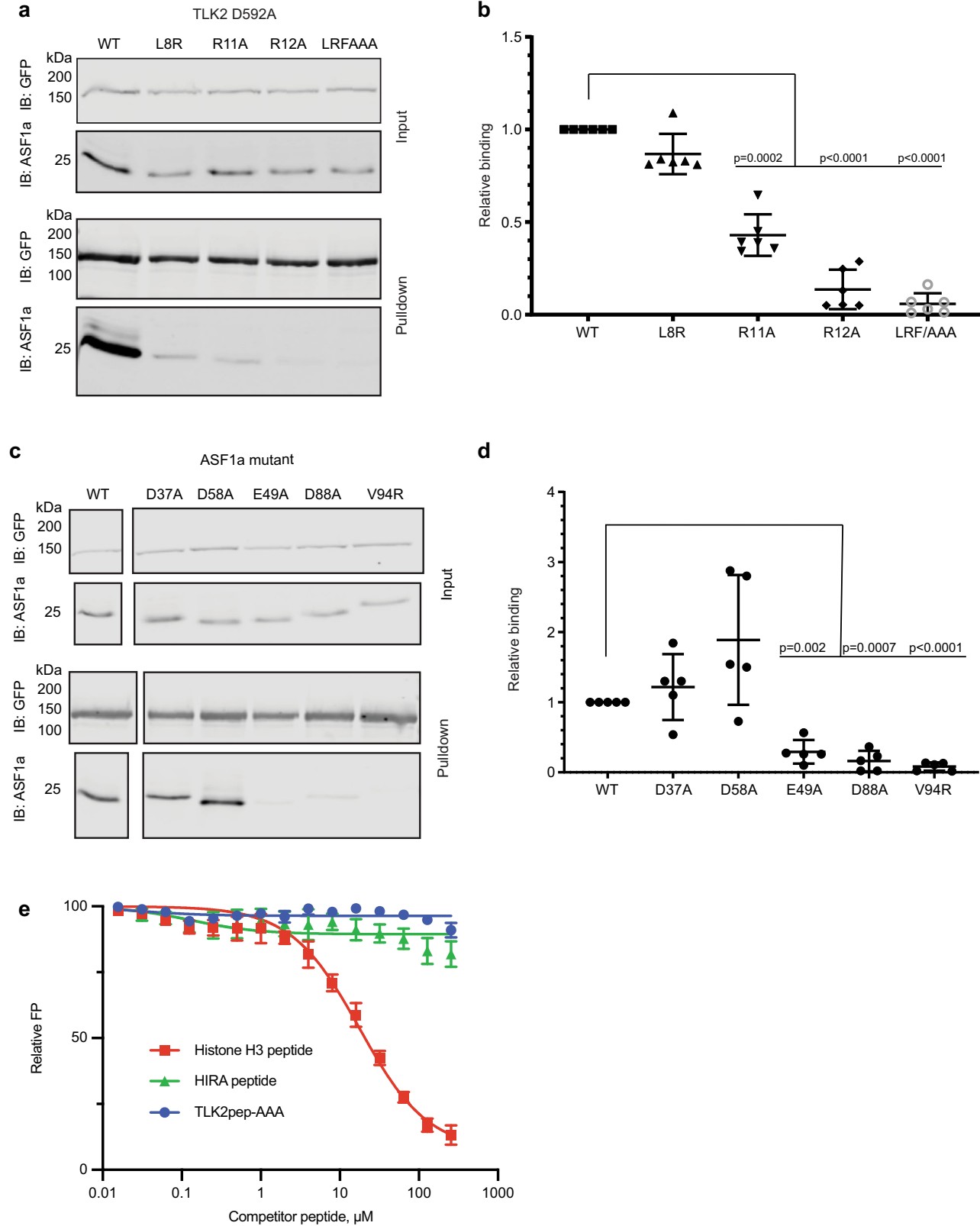

which they share a preference for acidic residues at the −2 position[65]. Notably, the catalytic cleft residue corresponding to TLK2-Lys557 is either Lys or Arg in all human PLK isoforms, potentially suggesting a common basis for phosphorylation site motif recognition.

We note that not all bona fide phosphorylation sites on ASF1a/b conform to a strict D/E-x-S consensus sequence (see Fig. 1c).

Thus, while catalytic site interactions are likely to influence the rate at which TLKs phosphorylate ASF1, they alone cannot dictate substrate selectivity, which likely depends on the non-catalytic docking interaction between the two proteins. Docking interactions are commonly used for substrate recruitment by kinases and other protein modification enzymes, and frequently occur through recognition of short linear sequence motifs

**Fig. 5 The H3-binding site is required for TLK2 binding to ASF1. a, b** Cells were co-transfected with FLAG-tagged ASF1a and GFP-tagged kinase-inactive TLK2 (D592A) or GFP-TLK2 (D592A) containing additional mutations at the structurally defined ASF1a-binding site and anti-GFP nanobody pulldown assays were performed. Precipitated GFP proteins and co-precipitating FLAG-ASF1a were fractionated by SDS-PAGE and detected by immunoblotting. A representative experiment is shown in **a**. Relative ASF1a binding, corrected for FLAG-ASF1a input and normalized to GFP-TLK2 (D592A) was quantified, and mean ± SD ($n = 6$ independent experiments) is shown in **b**. p values in one-way ANOVA with Dunnett's multiple comparison correction are indicated. **c, d** Pulldown assays were performed as in A from cells co-expressing GFP-TLK2 (D592A) and WT or mutant FLAG-tagged ASF1a. A representative experiment is shown in **c**. Relative ASF1a binding, normalized for FLAG-ASF1 input was quantified in five independent replicates and mean ± SD is shown in **d**. p values in one-way ANOVA with Dunnett's multiple comparison correction are indicated. **e** Competitive FP assays of peptide binding to ASF1a-NT were performed as in Fig. 3d. Peptides correspond to the ASF1-binding fragments of human histone H3.1 (120–135) and HIRA (446–464). TLK2pep-AAA is the TLK2 N-terminal peptide harboring L16A/R19A/F20A substitutions. Results are shown as mean ± SD ($n = 3$ biologically independent experiments). Source data are provided as a Source Data file.

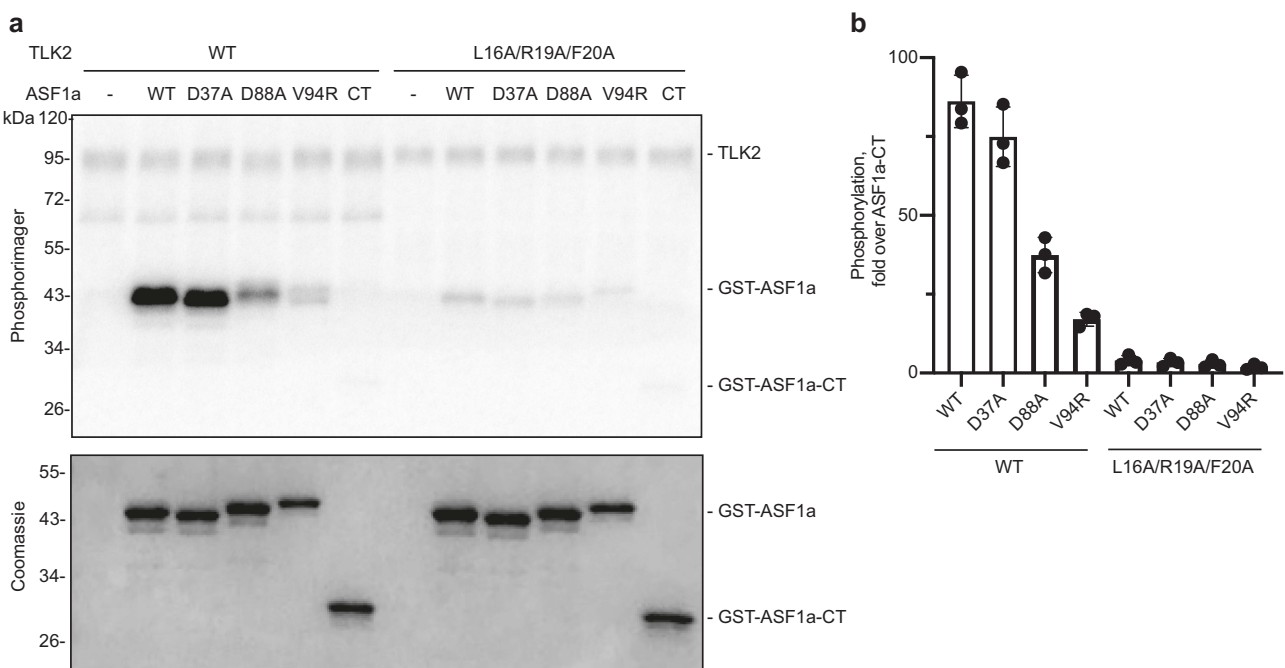

**Fig. 6 TLK2 binding at the H3-binding site is important for ASF1 phosphorylation. a, b** Purified GST tagged ASF1a WT, indicated mutants, or C-terminal tail residues 155–204 (CT) were phosphorylated by WT TLK2 or the indicated mutants with [γ-$^{32}$P]ATP. Samples were subjected to SDS-PAGE followed by phosphor imaging. A representative experiment is shown in **a**. Quantified radiolabel incorporation for each of three independent replicates (mean ± SD) is shown relative to the level of phosphorylation of ASF1a-CT by the indicated form of TLK2. Source data are provided as a Source Data file.

(SLiMs) present within substrates by either the kinase itself or an accessory adapter subunit or scaffold protein[66]. Our results show the TLK2-ASF1a interaction is unusual in that it involves a short sequence present within the kinase that interacts with a globular domain of its substrate. This arrangement could help to enforce specificity for a single target protein, as proteins other than ASF1 are unlikely to have a similar recognition domain. A similar phenomenon is involved in substrate targeting by MAP kinase kinases, which exclusively phosphorylate MAP kinases[67]. It is possible that binding to ASF1a/b can recruit TLKs to phosphorylate associated proteins, or that TLKs share a common interface with other histone H3–H4 binding proteins.

Mutagenesis of the histone-binding site of ASF1a has been shown to inhibit association with TLK1[48], however the nature of the interaction had not been explored. Our co-crystal structure revealed that the TLK2 N-terminal region strikingly recapitulates a fragment of histone H3 in its interaction with ASF1a, suggesting that TLKs recruit ASF1 by mimicking their chaperone client. In this way the evolution of ASF1 regulation may have been facilitated by the presence of a functionally important helix-interacting hydrophobic groove that could be exploited for other interactions. Another implication of this finding is that TLK2 and H3–H4 dimers will compete for binding to ASF1, which is consistent with

prior results showing that phosphorylation of ASF1 is enhanced by histone deficiency and that TLK1 binding to ASF1 containing a V94R mutation in its histone H3 binding site is reduced[48]. Thus, published data[48] and our results suggest that TLK2 binds to free ASF1, driving ASF1 phosphorylation and enhancing histone H3–H4 dimer binding. It is likely that TLK2 binds preferentially to non-phosphorylated ASF1, while H3–H4 favors phospho-ASF1, and this could explain our observation that ASF1 interacts more strongly with kinase-inactive TLK2 than with WT TLK2. Modulation of binding by phosphorylation may allow TLKs to compete against the high levels of H3–H4 for the pool of non-phosphorylated ASF1.

In addition to the histone H3 binding interface of ASF1 proteins, the presumably primordial binding sites for histone H4 and the HIRA co-chaperone have also been co-opted for interaction with regulatory proteins. For example, a multivalent interaction involving multiple interfaces has been described for binding of budding yeast Asf1 to the checkpoint kinase Rad53. A peptide from the C-terminus of Rad53 engages the Asf1 N-terminal domain making contacts at both the histone H4-binding pocket and the B-domain-binding site[68]. The Asf1 histone H3-binding site is also apparently involved in binding to Rad53, although the basis for this interaction remains unclear[68]. In addition,

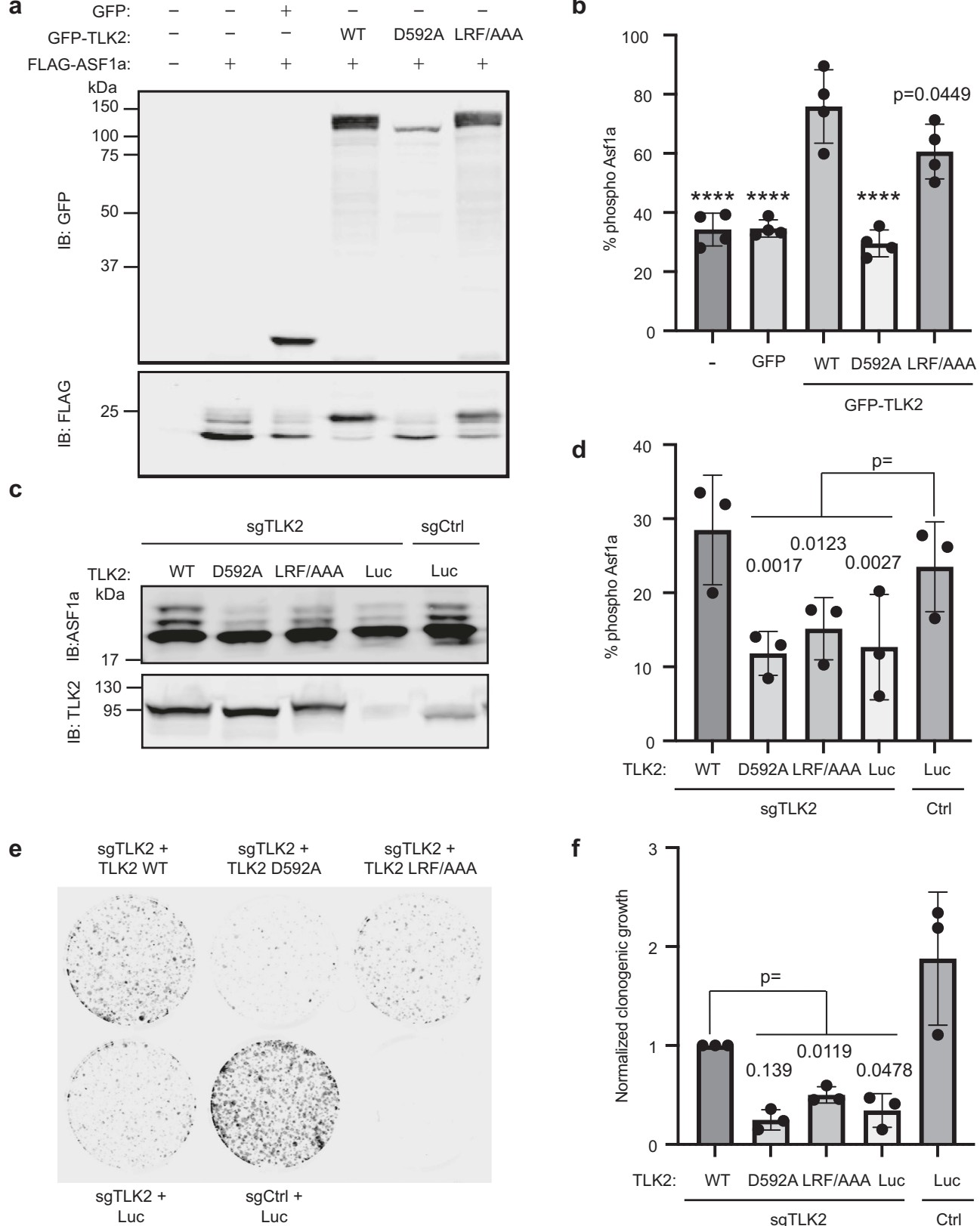

phosphorylation of a site in the Asf1 C-terminal tail region by casein kinase 2 induces association with the Rad53 FHA1 domain to further stabilize the interaction. In contrast to TLK2, where a multivalent association with ASF1a/b has a role in substrate recognition, Rad53 does not appear to phosphorylate Asf1, but rather delivers it to sites of DNA damage to coordinate DNA repair with chromatin assembly[12,14,69].

The observation that the N-terminus of TLK2 contacts ASF1 at its histone H3 binding site is interesting in light of recent efforts to therapeutically target ASF1. The link between histone

**Fig. 7 TLK2 binding to ASF1 promotes phosphorylation in cells and supports cell growth. a, b** FLAG-tagged ASF1a was transiently co-expressed in HEK 293T cells with GFP, or GFP-tagged WT or mutant TLK2. After 24 h, cells were lysed, fractionated by SDS-PAGE and immunoblotted with anti-GFP and anti-FLAG antibodies. The extent of FLAG-tagged ASF1a phosphorylation was calculated as the ratio of slower migrating species to total FLAG signal in four independent replicates, and mean ± SD is shown in **b**. Values significantly different from GFP-TLK2 in one-way ANOVA with Dunnett's multiple comparison test are indicated ****$p < 0.001$. **c, d** TLK2 and ASF1a were detected by immunoblotting lysates of control (Ctrl) or TLK2-knockout (sgTLK2) MCF7 cells that stably express WT or mutant TLK2 or luciferase (Luc) as a control. ASF1a phosphorylation was calculated by electrophoretic mobility shift in three independent replicates and mean ± SD is shown in **d**. $p$ values of significant differences from Ctrl + Luc value in one-way ANOVA with Dunnett's multiple comparison test are indicated. **e, f** The ability of control or TLK2 knockout cells expressing luciferase, WT, or mutant TLK2 to form colonies after 2 weeks of culture was assessed and quantified. Representative images of crystal violet-stained cells visualized with a LiCor Odyssey scanner are shown in **e** and quantification of three independent experiments normalized to the sgTLK2 + TLK2 WT value (mean ± SD) is shown in **f**. Values significantly different from sgTLK2 cells expressing WT TLK2 in one-way ANOVA with Dunnett's multiple comparison test are indicated. Source data are provided as a Source Data file.

chaperones and cancer is now well appreciated[2,70,71], and ASF1 inhibitors have been suggested as potential anticancer agents[72–74]. Several teams have sought to disrupt ASF1-histone interactions by targeting the histone-binding sites on ASF1[72–74]. While identification of small molecules capable of effectively disrupting the high-affinity ASF1–histone interaction is challenging, peptide-based inhibitors have shown efficacy in cultured cells and in a mouse xenograft model[73]. Our finding that TLK2 binds in a similar manner to histone H3 suggests that compounds interacting with the H3-binding site are also likely to prevent TLK2 binding to ASF1, hence inhibiting ASF1 activation and indirectly reducing histone binding. As the TLK2–ASF1 interaction is of lower affinity (~2 μM) than the histone-ASF1 interaction (sub-nM)[48], and TLK2 is likely to be present at lower concentrations than histone H3, small molecule inhibitors that effectively perturb TLK2 binding to ASF1 in cells might be more readily identified.

## Methods

**Cloning**. Mammalian expression constructs for FL human TLK2 and its N-terminal truncations were generated by PCR amplification from a FL TLK2 cDNA template (CCSB human ORFeome orf_ID #10519) and subcloning into pEGFP-C1 or pcDNA3-FLAG. The mammalian expression vector encoding 3×FLAG epitope-tagged FL human TLK2 and the lentiviral expression vectors encoding TLK2 WT and mutants were generated by Gateway recombination (ThermoFisher Scientific) into the pV1900 destination vector[75] and pLX304-V5-Blast vector (a gift from David Root, Addgene plasmid #25890), respectively. The pLX304-Luciferase-V5-Blast vector was a gift from Kevin Janes (Addgene plasmid #98580)[76]. C-terminal TLK2 truncations were generated by introducing stop codons using QuikChange mutagenesis. The untagged FL human ASF1a mammalian expression construct was generated by Gateway recombination from pDONR223-ASF1a (CCSB human ORFeome orf_ID #5980) into pV1900. Bacterial expression plasmids encoding FL human ASF1a and ASF1b and fragments thereof as N-terminal GST fusion proteins were generated by PCR amplification and cloned into pGEX4T-1. Structure-based point mutations of TLK2 and ASF1a were introduced by QuikChange mutagenesis. Phosphorylation site mutants of ASF1b were generated by Q5 mutagenesis (New England Biolabs). A CRISPR/Cas9 single guide targeting the human *TLK2* gene (target sequence GAGCCTCATTTACTG AACAG)[77] was inserted into LentiCRISPRv2-Puro vector (a gift from Brett Stringer, Addgene plasmid # 98290[78]). See Supplementary Table 2 for all primer sequences. All constructs used were authenticated by DNA sequencing through the entire ORF.

**Protein and peptide production**. C-terminally hexahistidine-tagged anti-GFP nanobody[54] cloned into pET-28a (a kind gift from Topher Carroll) was expressed in BL21 (DE3) RIPL *E. coli* and purified using the following protocol. LB media with kanamycin and chloramphenicol was inoculated with 1/200th volume of an overnight starter culture and was incubated at 200 rpm and 37 °C until it reached an OD$_{600}$ of 1.3. Cultures were cooled-down to 16 °C at 200 rpm for 2 h, induced with 0.1 mM isopropyl β-D-1-thiogalactopyranoside (IPTG), and incubated 18 h at 16 °C and 200 rpm. Pellets were resuspended in buffer A [20 mM HEPES, 250 mM NaCl, 10% glycerol, 20 mM imidazole, 1× cOmplete$^{TM}$ EDTA free protease inhibitor cocktail (Roche), 1 mg/ml lysozyme (AmericanBio), pH 7.5] and incubated 1 h at 4 °C. Cells were disrupted by sonication and centrifuged for 30 min at 22,000 × *g*. The supernatant was loaded on Ni-NTA agarose beads (Qiagen), washed with 15 column volumes of buffer B (20 mM HEPES, 250 mM NaCl, 10% glycerol, 20 mM imidazole, pH 7.5) and eluted with 10 column volumes of buffer C

(20 mM HEPES, 250 mM NaCl, 5% glycerol, 200 mM imidazole, pH 7.5). Eluate was dialyzed overnight at 4 °C in buffer D (1× PBS, 10% glycerol) and further purified on a HiLoad 16/60 S200 size exclusion column (GE Healthcare) equilibrated in buffer D. Collected fractions were diluted in buffer D to 0.2 mg/ml, flash frozen in liquid nitrogen and stored at −80 °C.

ASF1a-NT for crystallography, DSF and FP assays was expressed in BL21 (DE3) Rosetta 2 cells as follows. An overnight starter culture was diluted 1:200 in LB media with ampicillin and chloramphenicol and was incubated at 200 rpm and 37 °C until it reached an OD$_{600}$ of 0.7. After chilling to 16 °C at 200 rpm for 2 h, cultures were induced with 0.2 mM IPTG and incubated 20 h at 16 °C and 200 rpm. Pellets were resuspended in buffer A [50 mM Tris-HCl, 250 mM NaCl, 5% glycerol, 0.1 mM tris(2-carboxyethyl) phosphine (TCEP), 1× cOmplete$^{TM}$ protease inhibitor cocktail (Roche), 1 mg/ml lysozyme, pH 8.2] and incubated 1 h at 4 °C. Cells were disrupted by sonication and centrifuged for 30 min at 22,000 × *g*. The supernatant was loaded on glutathione-Sepharose 4B beads (GE Healthcare), washed with 50 column volumes of buffer B (50 mM Tris-HCl, 250 mM NaCl, 0.1 mM TCEP, pH 8.2) and eluted with ten column volumes of buffer C (50 mM Tris-HCl, 250 mM NaCl, 0.1 mM TCEP, 20 mM reduced glutathione, pH 8.2). Eluate was dialyzed overnight at 4 °C in buffer D (20 mM HEPES, 250 mM NaCl, 0.1 mM TCEP, pH 7.5) and further purified on a HiLoad 16/60 S200 size exclusion column (GE Healthcare) equilibrated in buffer D. Collected fractions were concentrated in buffer D to 14 mg/ml, flash frozen in liquid nitrogen and stored at −80 °C. Proteins used as substrates in kinases assays were expressed in BL21 (DE3) LysS cells, purified by GSH affinity chromatography as described above, and frozen down without further purification.

FL TLK2 WT and LRF/AAA mutant used for kinase assays were expressed and purified from *E. coli* BL21 (DE3) Rosetta 2 cells as follows: 3 × 1 l of LB media in 2 l flasks with ampicillin and chloramphenicol were inoculated with 1/200th volume of a saturated starter culture and incubated at 200 rpm and 37 °C until it reached an OD$_{600}$ of 1.1. Cultures were cooled to 16 °C at 200 rpm for 2 h, induced with 0.1 mM IPTG and incubated 20 h at 16 °C and 200 rpm. Bacteria were resuspended in buffer A [20 mM HEPES, 250 mM NaCl, 5% glycerol, 20 mM imidazole, 1× cOmpleteTM EDTA free protease inhibitor cocktail, 1 mg/ml lysozyme, 0.1% Triton X-100, 0.1 mg/ml DNase I (Roche), 0.2 mM TCEP, pH7.5] and incubated 1 h at 4 °C. Cells were disrupted by sonication and centrifuged for 60 min at 48,000 × *g*. The supernatant was loaded onto Ni-NTA agarose beads (Qiagen), which were washed with 10 column volumes of buffer B (20 mM HEPES, 250 mM NaCl, 20 mM imidazole, 0.1 mM TCEP, pH 7.5), 50 column volumes of buffer C (20 mM HEPES, 500 mM NaCl, 20 mM imidazole, 5% glycerol, 0.1 mM TCEP, pH 7.5) and 10 column volumes of buffer B. Bound proteins were eluted with 10 column volumes of buffer E (20 mM HEPES, 250 mM NaCl, 300 mM imidazole, 0.1 mM TCEP, pH 7.5). Eluate was concentrated in a 30 kDa MWCO centricon (Millipore) centrifugal filter and further purified on a HiLoad 16/60 S200 size exclusion column (GE Healthcare) equilibrated in buffer F (20 mM HEPES, 250 mM NaCl, 50 mM imidazole, 0.1 mM TCEP, pH 7.5). Collected fractions were assessed for purity by SDS-PAGE and Coomassie staining, flash frozen in liquid N$_2$ and stored at −80 °C.

FLAG epitope-tagged TLK2 ΔN178 and FL TLK2 (WT and kinase inactive D592A mutant) used for the assays shown in Fig. 1 were prepared by transient expression of the encoding plasmids using linear polyethylenimine (PEI, MW 25,000) (Polysciences, Inc.; Warrington, PA) in HEK293T cells in 15 cm dishes. After 40–44 h, cells were washed in ice-cold PBS, and lysed in 2.25 mL per plate of 20 mM Tris, 150 mM NaCl, 1 mM EDTA, 1 mM EGTA, 1% Triton X-100, 2.5 mM sodium pyrophosphate, 1 mM β-glycerophosphate, 1 mM Na$_3$VO$_4$, 1 mM DTT, 1 mM PMSF, 10 μg/ml leupeptin, 2 μg/ml peptstatin A, 10 μg/ml aprotinin, pH 7.5. Following incubation on ice for 10 min, lysates were cleared by centrifugation and tumbled with 32.5 μl of anti-FLAG M2 affinity gel (Millipore-Sigma #A2220) for 2 h. Beads were pelleted, washed twice with lysis buffer, and twice with kinase elution buffer (50 mM HEPES, 100 mM NaCl, 1 mM DTT, 5 mM β-glycerophosphate, 0.1 mM Na$_3$VO$_4$, 0.01% Igepal CA630, 10% glycerol, pH 7.4). Protein was eluted by incubation with 65 μl kinase elution buffer containing 0.5 mg/ml 3× FLAG peptide (Millipore-Sigma #F4799) for 1 h.

After centrifugation, the supernatant was filtered through a Whatman Unifilter 800 plate, snap frozen in a dry ice-ethanol slurry and stored at −80 °C.

The fluorescein-labeled TLK2 tracer peptide for FP experiments was synthesized at Tufts University Core Facility and purified by HPLC. All other peptides were custom synthesized by Genscript at >98% purity (TLK2, histone H3 and HIRA peptides) or >95% purity (kinase assay substrates). Peptide sequences were: TLK2, EELHSLDPRRQELLEARFTGV; TLK2-AAA, EELHSLDPRRQEL AEAAATGV; Histone H3, TIMPKDIQLARRIRGERA; HIRA, LKKQVETRTA DGRRRITPL. TLK2 peptide used for crystallography was dissolved in 20 mM HEPES, 250 mM NaCl, 0.1 mM TCEP, pH 7.5 at 26 mM final concentration, and stored at −20 °C.

**Crystallization, X-ray data collection, and structure determination**. The human ASF1a apo-structure (PDB accession code 7LNY) was generated by crystallizing 14 mg/ml ASF1a-NT in hanging drops in VDX 24-well plates (Hampton Research) with a 1 ml reservoir of 22% PEG3350 and 6% of Tascimate pH 6.0. One microliter of ASF1a was mixed with 1 µl of mother liquor and incubated 29 days at 20 °C. Crystals were soaked in 50% 2 M sodium malonate, 10% glycerol, and 50% mother liquor for 1 min before freezing in liquid nitrogen. For the ASF1a·TLK2 peptide complex structure (PDB accession code 7LO0), 125 µl of ASF1a-NT at 14 mg/ml was mixed with 43 µl of 26 mM hTLK2 peptide and crystallized in hanging drops in VDX™ 24-well plates (Hampton Research) with a 1 ml reservoir of 18% PEG3350, 0.2 M of trisodium citrate, pH 5.6 and 2% octyl-D-glucose. ASF1a-NT (1 µl) was mixed with 1 µl of mother liquor and incubated 34 days at 20 °C. Crystals were soaked in 25% PEG200 and 75% mother liquor for 1 min before freezing in liquid nitrogen. X-ray data were collected at 0.9792 Å and 100 K on NE-CAT beamline 24ID-C at the Advanced Photon Source (APS), Argonne National Laboratory. X-ray data were integrated and scaled using XDS[79] and Aimless[80]. Phases were determined by the molecular replacement method using available ASF1a (chain A residues 1–154) coordinates[6] as a search model with Phenix-PHASER[81]. Initial reconstruction was done with Phenix Autobuild[82] and several rounds of manual building were performed with COOT[83] and structure refinement with Phenix Refine[58]. MOLPROBITY[84] was used for model validation. In the ASF1a·TLK2 peptide complex structure, 99.8% of protein residues were Ramachandran favored and 1 residue (0.075%) was an outlier. In the case of the Asf1a apo-structure, 98.2% of protein residues were Ramachandran favored and 0% were outliers. Crystallographic X-ray data and structure refinement statistics are summarized in Table 1. All figures were prepared with Pymol (Schrodinger, Version 2.4).

**Protein–protein interaction pull-down experiments**. NHS-activated Sepharose 4 fastflow beads (GE Healthcare) were crosslinked to GFP nanobodies according to the manufacturer's protocol. In brief, 1.25 ml of anti-GFP nanobody at 0.2 mg/ml were mixed with 0.25 ml of 1 M NaHCO₃, pH 8.3 and centrifuged 15 min at 24,000 × g and 4 °C. Beads (2.5 ml) were washed with 3 × 12 ml of 1 mM HCl. Anti-GFP nanobody was mixed with the beads and incubated 24 h at 4 °C with rotation. The beads were then incubated 4 h at room temperature with 2 ml of 0.1 M Tris pH 8.5 and washed three times with 10 ml of 0.1 M Tris, pH 8.5. Beads were washed three more times with 10 ml of 0.1 M NaOAc, 0.5 M NaCl, pH 4.5 and three times with 10 ml of 1× PBS. For storage at 4 °C, the beads were resuspended in 2.5 ml of 1× PBS and 1× cOmplete™ EDTA free protease inhibitor cocktail (Roche).

HEK 293T cells were cultured at 37 °C in 5% CO₂ in Dulbecco's modified Eagle's medium (DMEM) containing 9% fetal bovine serum, 1% sodium pyruvate, and 1% nonessential amino acids (Gibco). For transfection, cells cultured in 10 cm dishes were incubated for 30 min at 37 °C and 5% CO₂ in DMEM with 2 µg of DNA and 6 µl of 1 mg/ml of PEI for each plasmid. Cells were subsequently incubated for 48 h at 37 °C and 5% CO₂ in complete DMEM as described above. Cells were lysed using 1 ml of buffer XT (50 mM sodium fluoride, 40 mM sodium pyrophosphate, 50 mM NaCl, 150 mM sucrose, 10 mM PIPES, 0.1% Triton X-100, 2 mM Na₃VO₄, 1× cOmplete™ protease inhibitor cocktail, pH 6.8) and incubated 2 h at 4 °C with 50 µl of anti-GFP nanobody beads suspension. Beads were washed three times with 1 ml of buffer XT and separated by SDS-PAGE (12% acrylamide) after resuspension in 5× Laemmli sample buffer and heating for 5 min at 95 °C. Samples were transferred to 0.45 µm nitrocellulose membranes, which were blocked for 30 min in 5% non-fat milk in TBS-T. Immunoblotting was performed with primary antibodies diluted in 5% non-fat milk in TBS-T [1:1000 of anti-GFP (Rockland, catalog #600-101-215), anti-FLAG (Millipore-Sigma, #F1804) or anti-ASF1a (Cell Signaling Technology, C6E10 Rabbit mAb #2990)] and with secondary antibodies conjugated to fluorescent IR Dye800 or IR Dye680 were purchased from LI-COR Biosciences and used at 1:25,000 dilution in 5% non-fat milk in TBS-T (IRDye 680RD goat anti-mouse IgG, catalog #92668070; IRDye 800CW donkey anti-mouse IgG, catalog #92632212; IRDye 680RD donkey anti-rabbit IgG, catalog #92632213; IRDye 800CW donkey anti-rabbit IgG, catalog #926680723; IRDye 680RD donkey anti-goat IgG, catalog #92668074) and scanned on the Odyssey CLx infrared imaging system. Quantification was done with LiCor Image Studio Lite v5.2 and statistical analysis was conducted using the one-way ANOVA function of Graphpad Prism version 9.3.0.

**Kinase assays**. For PSPL analysis[75], kinase assay buffer (2 µl 50 mM Tris, pH 7.5, 100 mM NaCl, 10 mM MgCl₂, 1 mM DTT) containing 0.1% Tween 20 was added to wells of a 1536-well plate using a Mantis nanodispening liquid handler (Formulatrix). The PSPL (Anaspec) was transferred to reaction plates in 200 nl aliquots from stock plates (0.6 mM in ddH₂O) manually by pin tool (V&P Scientific). Following addition of kinase (to 8 ng/µl) and [γ-³³P]ATP (to 50 µM, 15 nCi/µl), plates were sealed, incubated at 30 °C for 2 h, chilled on ice, and unsealed. Aliquots (200 nl) of each well were pinned onto streptavidin membrane (Promega), which was washed twice for 3 min with 0.1% SDS in TBS, twice with 2 M NaCl, and twice with 2 M NaCl containing 1% H₃PO₄, and rinsed twice briefly with ddH₂O. Membranes were air-dried and analyzed by phosphor imaging. Normalized quantified spot intensities from two separate experiments were averaged, and log2 transformed data were used to generate heat maps (Microsoft Excel) or sequence logos (Seq2Logo).

For assays of ASF1a/b phosphorylation, TLK2 and 2 µM substrate were incubated in kinase assay buffer containing 0.02% Igepal CA630 and 50 µM [γ-³²P] ATP (0.1 µCi/µl) for 15 min at 30 °C. Final kinase concentrations used were 3 nM for HEK293T cell-expressed TLK2 ΔN178, 0.2 nM for HEK293T cell-expressed FL TLK2 in Fig. 1, and 1.2 nM for bacterially-expressed FL TLK2 in Fig. 6. Reactions were quenched with SDS-PAGE loading buffer, heated to 95 °C for 5 min, and separated by SDS-PAGE (10% acrylamide). Gels were stained with Coomassie, dried, and subjected to phosphor imaging. Radiolabel incorporation was quantified using ImageJ. Synthetic peptide kinase assays were conducted similarly, except that reactions were carried out for 30 min, contained 10 µM substrate and 1.5 nM TLK2 ΔN178, and experiments were performed in technical duplicate. Aliquots were withdrawn every 10 min and spotted onto P81 filters (Whatman), which were immersed in 75 mM H₃PO₄. Filters were washed three times with 75 mM H₃PO₄ (5 min per wash), rinsed with acetone, air-dried, and radiolabel incorporation determined by scintillation counting.

**Lentivirus production**. Lentiviruses were produced by co-transfecting HEK293T cells (ATCC catalog # CRL-3216) with packaging vectors psPAX2 (viral proteins Gag and Rev under the SV40 promoter; Addgene plasmid #12260, a gift from D. Trono) and pMD2.G (viral protein VSV-G expressed under the CMV promoter; Addgene plasmid #12259, a gift from D. Trono) together with the LentiCRISPRv2 or pLX304-V5 constructs described above. Viral supernatant was collected 48 h and 72 h after transfection and filtered with a 0.45 µm filter. Cell lines were transduced by incubation with viral supernatant (neat or diluted) and 8 µg/ml polybrene (Millipore-Sigma) for 18 h, and stable lines were established by antibiotic selection.

**Analysis of ASF1a phosphorylation in cells**. MCF7 cells (ATCC catalog # HTB-22) were cultured at 37 °C in 5% CO₂ in Eagle's minimum essential medium (EMEM, ATCC) containing 9% fetal bovine serum and 0.01 mg/ml insulin (Santa Cruz Biotechnology). Cells stably expressing control luciferase, or C-terminally V5-tagged WT or mutant TLK2 were generated by lentiviral infection followed by selection with 7.5 µg/ml blasticidin for 10 days. Cells were then infected with control (no sg insert) or TLK2-targeting CRISPR/Cas9 lentivirus particles and selected with 3 µg/ml puromycin for 2 days. To analyze ASF1a phosphorylation, cells were seeded at 25% density in 6-well dishes and cultured overnight. Cells were washed once with PBS and lysed in 100 µl SDS-PAGE sample buffer lacking β-mercaptoethanol. Lysates were passed through a 23G needle six times, and β-mercaptoethanol was added to a final concentration of 2.5%, and samples were heated to boiling for 5 min. Samples were fractionated by SDS-PAGE (12% acrylamide) and transferred to PVDF membranes, which were probed with antibodies to ASF1a and TLK2 (Santa Cruz, sc-393506) diluted 1:1000 in 5% non-fat milk in TBS-T. The extent of ASF1a phosphorylation was calculated as the ratio of slower migrating species to total protein as quantified using ImageJ.

**Clonogenic assays**. MCF7 cells were detached with trypsin/EDTA, resuspended in complete medium, passed through a nylon mesh filter, and counted. Cells were seeded at 10,000 cells per well in 6-well plates and cultured in complete medium for 2 weeks. After incubation, cells were gently washed with PBS, stained with PBS containing 0.5% crystal violet, 6% formaldehyde and 1% methanol for 15 min, washed three times with water and air dried. The total signal in each well was quantified with a LiCor Odyssey scanner using the 700 nm channel.

**Fluorescence polarization**. ASF1a 1–155 (63 nM) was mixed with tracer peptide (5 nM, FITC-βAla-LHSLDPRRQELLEARFTG) in the presence or absence of varying concentrations of unlabeled TLK2 peptide (two-fold increments ranging from 15.6 nM to 256 µM) in 40 µl FP buffer (50 mM Tris pH 8.0, 100 mM NaCl, 10 mM MgCl₂, 1.2 mM DTT, 0.012% Brij-35, 0.2 mg/ml BSA) in duplicate in a 384-well plate. Samples with tracer peptide alone were included as a minimum FP control. Plates were incubated at room temperature for 2 min with gentle shaking and centrifuged 5 min at 4000 rpm. Plates were read in a Tecan Infinite M1000 Pro instrument in FP mode at 295 K with excitation wavelength = 470 nm and emission wavelength = 520 nm. Reads were normalized to the zero and maximum FP signal controls. The experiment was performed three times, and data were fitted by three parameter non-linear regression using GraphPad Prism version 9.3.0.

**Differential scanning fluorimetry**. Each well of a BioRad MLL9601 plate was filled with 18 µl of DSF buffer (20 mM HEPES, 250 mM NaCl, 0.1 mM TCEP, pH 7.5), 2 µl of ASF1a 1–155 at 100 µM, 2 µl of TLK2 peptide at 2 mM or 2 µl of DSF buffer, and 3 µl of 5000 X SYPRO$^{TM}$ orange (Invitrogen) at 62×. Denaturation was measured on a Bio-Rad qPCR machine with initial temperature stabilization at 4 °C for 5 min followed by a ramp from 4 to 95 °C at 1 °C/min and imaging on FAM filter every 1 °C. Paired $t$-test statistical analysis was conducted using Graphpad Prism 8.2.1.

**Sequence and structure analysis**. Sequences of human histone 3.1 (P68431), TLK1 (human: Q9UKI8, mouse: Q8C0V0, zebrafish: Q90ZY6, chicken: F1P396) and TLK2 (human: Q86UE8, mouse: O55047, zebrafish: Q1ECX4, chicken: A0A3Q2TUS7) were collected on the UniProt database[85], aligned using T-coffee[86], and rendered with ESPript 3.0 online software[87]. For alignments of full-length TLKs, protein sequences of TLK1 and TLK2 from a range of animal species were initially aligned using Multalign, imported into Jalview, and then re-aligned with Clustal via the JABWAS server[88,89].

Root mean square deviations between structures were determined using the protein structure comparison service PDBeFold at European Bioinformatics Institute[13]. Interface buried areas, hydrogen bonds, and salt bridges were determined using Protein interfaces, surfaces and assemblies service PISA at the European Bioinformatics Institute[90].

**Reporting summary**. Further information on research design is available in the Nature Research Reporting Summary linked to this article.

## Data availability

Structural coordinates were deposited in the Protein Data Bank (www.wwpdb.org) under the accession codes PDB 7LNY (Apo-ASF1a) and PDB 7LO0 (ASF1a/TLK2 peptide complex). Previously published structures used were obtained from the Protein Data Bank: 5O0Y (TLK2 kinase domain), 1ATP (catalytic subunit of cAMP-dependent protein kinase), 2I32 (ASF1a-HIRA complex) and 2IO5 (CIA-histone H3-H4 complex). Sequences were obtained from the Uniprot database: P68431 (human histone 3.1), Q9UKI8 (human TLK1), Q8C0V0 (mouse TLK1), Q90ZY6 (zebrafish TLK1), F1P396 (chicken TLK1), Q86UE8 (human TLK2), O55047 (mouse TLK2), Q1ECX4 (zebrafish TLK2), A0A3Q2TUS7 (chicken TLK2). Constructs and reagents are available from the authors upon request. Source data are provided with this paper.

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

## Acknowledgements

This work was supported by NIH R03 TR002912, the Yale SPORE in Lung Cancer (5P50 CA196530) and the Neuwirth Beatrice Kleinberg Research Gift to B.E.T. and D.A.C. B.S. was supported by AHA Postdoctoral Fellowship 18POST33960340. This research used resources of the Advanced Photon Source, a U.S. Department of Energy (DOE) Office of Science User Facility operated for the DOE Office of Science by Argonne National Laboratory under Contract No. DE-AC02-06CH11357.

## Author contributions

B.S., B.E.T., and D.A.C conceived the project, interpreted the results and wrote the manuscript. B.S., H.J.L., C.H-C., and B.G.S. performed the experiments and interpreted results. T.J.B. advised B.S. on structure determination. All authors edited and approved the submitted manuscript.

## Competing interests

The authors declare no competing interests.
