## [Peer Review File · Nature Communications]

REVIEWER COMMENTS

Reviewer #1 (Remarks to the Author):

The authors examine the mechanism of TLK2 recognition of one of its substrates, Asf1. Using the kinase domain of TLK2 (TLK2 deltaN178) in an analysis of kinase activity with an arrayed positional scanning peptide library, the authors generate heat maps and a sequence logo that reveal an intrinsic selectivity for the Asf1 amino acids adjacent to a phosphorylation site. Experimental data support this analysis. Further, the authors observe higher levels of phosphorylation by TLK2 when FL Asf1, compared to Asf1-CT is a substrate, pointing to an additional site of interaction with TLK2 within the globular core of Asf1. The deletion analyses and pull-down experiments in Figure 2 reveal the Asf1 binding region of TLK2 lies within 1-122, outside of the catalytic domain. This region was further refined to a peptide 2-22 of TLK1 in Figure 3. Biophysical approaches were used to show that TLK2 2-22 binds to the globular domain of Asf1 (1-155). Figure 4 shows the results of the crystal structure of Asf1a-NT-TLK2-NT peptide. The TLK2 peptide appears to bind as an alpha helix in the concave surface of Asf1 in a similar manner to helix 3 of H3, and also to the site where B-domain binding has been shown to occur. However, the authors show data and make good arguments for the second site being an artifact. Figure 5 addresses the relevance of the observed TLK2 containing residues and somewhat validates the observed interactions. Figure 6 tests the effects of TLK2 and Asf1 mutants on the level of phosphorylation of Asf1 and the results are as would be expected. Overall, the manuscript shows that there are two interaction surfaces between Asf1 and TLK2, one that involves the catalytic domain of TLK2 and the Asf1 C-terminal tail phosphorylation site and the other containing the N-terminal region of TLK2 and the H3 binding site within the globular core of Asf1.

Overall, this manuscript presents interesting new insights into the mechanism of Asf1 phosphorylation by TLK2. However, it does not go further to address quantitative aspects of the interactions or what the functional impacts of these findings are. There are a few issues to address, of which the most concerning ones relate to the crystallography, missing peptide controls, as well as clarity and presentation of several of the Figures.

Major:

1) With regard to the hypothesis generated by the major finding in the manuscript, that H3 and TLK2 bind in a mutually exclusive manner to Asf1, how does this impact the function in DNA repair or replication? There were no experiments addressing the functional implications in cells.

2) Crystallography: Validation report for the Asf1-TLK2 complex has unacceptable Ramachandran outliers. For a resolution of 2.71 Å, the electron density appears rather weak; at least the electron density in Supplemental Fig 3 is very hard to see. As such, the quality of the maps is hard to judge. It appears that the TLK2 helix (panel A) is very disordered compared to the beta-strand-like structure in the artifactual peptide (panel B).

3) Given that the helix is proposed, and the tested TLK2 mutants were only somewhat informative (it was necessary to make a triple mutant in order to show a loss of the interaction), the validation (Figure 5) could have benefited from using a disruptive mutation on the interacting surface of TLK2. Further, the TLK2 peptide was not tested to determine whether it adopts alpha helical conformation in the presence and absence of Asf1. Moreover, do the TLK2 amino acid substitutions alter the helicity of the peptides? It is important to know that the loss of binding is due to lost interactions and not due to disruption of the peptides' structure. Circular dichroism would be able to demonstrate that 1) the TLK2 peptide exists as a helix in solution, 2) whether the helical structure is induced by binding to Asf1 in the complex and 3) the mutants did not perturb the peptide structure.

4) Given the interesting structure of Asf1 with the alpha helical TLK2 peptide (9-21), little description and few informative figures were provided. In fact, the artifactual site had similar attention. For example, the structural comparisons are limited to the Asf1 protein and no superposition of the Asf1-H3 compared to the Asf21-TLK2 structure to illustrate the similarity of positions of the helices or key interacting residues was presented. It would be useful to determine how well conserved the positions are and whether there are any interactions that might be useful to discriminate between the binding interactions between Asf1 and of histones or TLK2. These might be the sites of informative mutants to address point 1 above.

5) The pHs and ionic strengths of the buffers used in different experiments is not the same and covers pH 7.5-8.0 and 100 mM to 250 mM salt, hampering direct or any quantitative comparisons.

6) Figures 1 use Asf1b, but the structural analyses were done with Asf1a. Is there a reason Asf1a wasn't used for all of the studies for consistency?

Minor:

1) There is no specific "histone chaperone domain" of Asf1 as mentioned lines 40, 50, 86-87, 159-160, 183, 186, 808 and 821 and Figure 1F. Asf1 is a histone chaperone and both the N-terminal

globular core domain and the intrinsically disordered C-terminal tail have been shown to contribute to histone binding, which is the primary activity of a histone chaperone. Asf1-NT or globular domain or core domain would all be appropriate alternatives.

2) KD is not defined in Figure 1. Also, assuming it refers to kinase dead, why is the KD still active? Is Asf1a1-155 the same as Asf1a-NT? The amino acids in the constructs are not described in the materials and methods or the Figure legends, only in the text. This makes it difficult to interpret the figures. What is the numbering of the amino acids noted in the TLK2 LRF/AAA mutant. There is also mention of a TLK2-D292A mutant in Figure 1 - is this the KD? It would improve clarity if these could be defined either in the methods or the Figure legends where they are used in addition to the text.

3) Figures: The canonical coloration of the histone proteins is blue for H3 and green for H4. The coloration used here is different for the same proteins in different figures and even in different panels of the same figure. For example, In Fig 1 Asf1 is green. In Figure 4 Asf1 is grey, TLK2 is orange and H3 is cyan and H4 yellow. In Supplemental Fig 3, TLK2 is cyan (panel A and B), Asf1 is yellow green and orange, Asf1 is grey and TLK2 is orange (panel D). In Supplemental Fig 4, Asf1 is grey, TLK2 is pink and HIRA is green (panel A and B), in Panel C Asf1 is in grey and cyan and TLK2 is orange. This inconsistency makes it hard to discern the identities of the proteins by looking at the figure. Canonical colors ought to be used for the histones and consistent (different) colors be used for the other proteins.

5) Line 194 – “The ASF1a structure appeared stable.” Crystallography cannot inform on protein stability.

Reviewer #2 (Remarks to the Author):

This study focuses on biochemical & structural characterizations of human Tousled-like kinase 2 (TLK2), which has been implicated in cooperating with histone chaperone Asf1 in cell cycle progression and maintenance of genome stability. By amino acid positional scanning, the authors found that the preferred TLK2 phosphorylation motif resembles those found at the C-terminal ends of Asf1a & Asf1b. They also found that TLK2 phosphorylates full-length Asf1 more efficiently than the C-terminal tail of Asf1, prompting them to identify an N-terminal short sequence that interact with the chaperone domain of Asf1. Biophysical and structural approaches show that they interact at an EC50 of ~2 μ M, and the TLK2 fragment forms an alpha helix and mimics the binding of histone H3 to Asf1. Structure guided mutagenesis, binding and in vitro phosphorylation assays largely confirmed

the structural findings, and the authors proposed that a bipartite recognition of Asf1 through its chaperone domain and the phosphorylation motif by the N-terminal helix and the kinase domain determines the substrate specificity of TLK2.

Overall, the manuscript is a clearly written, the data are decent, and the conclusions are justified. I recommend publication with minor changes.

Minor points:

1. Fig. 2A, top panel: the D592A pulldown has two bands, the lower band is stronger, is it ASF1 degradation? Please explain.
2. Fig. 2B: Quantitation shows D592A interacted stronger, is the lower band counted as Asf1?
3. Methods, lines 376, 392 & 411: The E. coli expression strain should be BL21 (DE3) with RIPL or Rosetta, to reflect the incorporation of the T7 expression system.

Reviewer #3 (Remarks to the Author):

The manuscript 'Tousled-like kinase 2 targets ASF1 histone chaperones through client mimicry' describes the mechanistic basis for TLK2 substrate specificity for ASF1a and ASF1b.

The authors provide molecular details of substrate selectivity and propose a model of substrate selection based on H3 mimicry. This conclusion is supported by solid structural and biochemical data. However, the concept that TLKs bind and phosphorylate histone-free ASF1 by binding to the histone-binding pocket of ASF1 was proposed previously (see below). The current work provides the structural basis for this model. However, the new mechanistic insights are not explored in terms of their implications for TLK function in cells. The structural insights open new opportunities for testing the model in cells, e.g. by replacing endogenous TLKs with mutants lacking the ASF1 binding motifs. Such experiments would be critical to expand the scope of this work.

The manuscript would also benefit from introducing the previous model at an early stage (e.g. introduction) and presenting their results in this context rather than presenting the interaction between TLK2 and the ASF1 histone binding pocket as a surprise. Moreover, previous work showing

an interaction between Rad53 and the histone binding pocket of scASf1 should also be introduced and discussed (Jiao et al., 2012).

Specific comments

1.

In the first part, the authors explore the TLK2 phosphorylation site motif and conclude that it has an unusual preference for acidic residues upstream of the phosphorylation site. The predictions from the motif analysis are tested on a single known phosphorylation site in Asf1b. This seems inadequate to make general conclusions and it is thus unclear whether the suggested motif is predictive for new TLK sites and substrates.

It would also be important to explore further the importance of the -1/-2 acidic residues on additional peptides. It is not entirely clear from the analysis that the acidic residues are more important than other residues. Would introduction of additional acidic residue on less strong sites improve phosphorylation?

2.

In the second part the authors identify an interaction between TLK2 and the ASf1b globular domain by apparent serendipity. However, this interaction was described many years ago by Klimovskaia et al., 2014 and led to the model that TLKs targets histone-free ASF1 for phosphorylation. The current work supports this model and provides the underlying structural basis for recognition of histone-free ASF1 by TLKs. This is interesting and important. However, the authors should demonstrate that this mechanism of H3 mimicry is important for TLK function and ASF1 phosphorylation in cells, e.g. by introducing truncated TLK2 lacking aa 2-23 into cells preferentially by genome editing or in a complementation system.

3.

The authors do not discuss previous work showing that the yeast kinase Rad53 binds to several sites on scASF1 including the histone H3 binding surface and the B-domain binding sites (Jiao et al., 2012). Given the substantial similarity with the findings in the current work, it is unclear why this literature is left out.

Minor points:

Figure 1F.

There is residual activity in the TLK2 KD samples. Is the kinase purified from cells? in which case there is probably endogenous wt kinase present due to oligomerization of TLK1/2 endogenous and exogenous.

Figure 2.

Confirm that the peptide 2-23 is responsible for the major binding activity by comparing 34-772 with 1-772.

Figure 3.

Test whether H3.1 and B-domain peptides can act competitors. The prediction is that only the H3.1 peptide should work.

Detailed response to reviewers

Reviewer #1

We thank the reviewer for their constructive comments and suggestions. We appreciate their opinion that “*this manuscript presents interesting new insights into the mechanism of Asf1 phosphorylation by TLK2*”. The reviewer identified a few issues to address, “*of which the most concerning ones relate to the crystallography, missing peptide controls, as well as clarity and presentation of several of the Figures*”. As detailed below, to address these concerns we have performed additional experiments and analyses and revised the figures and text of the resubmitted manuscript.

Major concerns:

- 1) “*With regard to the hypothesis generated by the major finding in the manuscript, that H3 and TLK2 bind in a mutually exclusive manner to Asf1, how does this impact the function in DNA repair or replication? There were no experiments addressing the functional implications in cells.*”

As the importance of Asf1 phosphorylation by TLKs is already well established in the literature (e.g., ¹⁻³), in this manuscript we focused on mechanisms regulating specificity of TLK2 phosphorylation of Asf1. Our results revealed the nature of the Asf1-TLK2 interaction, showed that H3 and TLK2 bind Asf1 in a mutually exclusive manner and established that perturbation of the interaction impaired TLK2-mediated Asf1 phosphorylation. However, as both this reviewer and Reviewer #3 requested cell-based experiments we assessed the consequences of CRISPR/Cas9-mediated loss of TLK2 and reconstitution with wild-type or mutant TLK2 on Asf1 phosphorylation and clonogenic growth in MCF7 breast cancer cells. Consistent with prior reports¹⁻³ we see that loss of TLK2 impairs Asf1 phosphorylation, assessed by gel shift on SDS-PAGE, and cell growth, assessed in colony formation assays. Notably, we show that reconstitution with wild-type TLK2 at least partially restores Asf1 phosphorylation and clonogenic growth, but reconstitution with the kinase inactive D592A mutant, or with an Asf1-binding defective L16A/R19A/F20A TLK2 mutant impairs this rescue. Thus, the intact Asf1-binding site in TLK2 is important for TLK2’s ability to phosphorylate AFS1 in cells and for cell growth. These results are described in the new **Fig 7** and on lines 316-339.

- 2) 2a) *Crystallography: Validation report for the Asf1-TLK2 complex has unacceptable Ramachandran outliers.*

We thank the reviewer for their careful assessment of the structure, which as they correctly point out is at 2.71Å. We note that the structure has a total of 1353 built amino acids. A single amino acid is in an outlier orientation in this structure (0.075% of the residues). We feel this is acceptable error at 2.71Å, particularly as this proline residue (Pro15 in chain B, shown in green) is very close to the orientation of the same residue the other seven copies of ASF1 (shown in yellow).

2b) For a resolution of 2.71 Å, the electron density appears rather weak; at least the electron density in Supplemental Fig 3 is very hard to see. As such, the quality of the maps is hard to judge. It appears that the TLK2 helix (panel A) is very disordered compared to the beta-strand-like structure in the artifactual peptide (panel B).

We thank the reviewer for pointing this out. We realize that for the initial submission we had only included the simulated annealing composite omit maps for the binding of the TLK2 helix, and had not included 2Fo-Fc and Fo-Fc maps. We have included two further panels for Supplementary Figure 3, panel C shows the 2Fo-Fc and Fo-Fc maps corresponding to panel A, and panel D shows the 2Fo-Fc and Fo-Fc maps corresponding to panel B. 2Fo-Fc maps are colored blue and contoured at 1 sigma, and Fo-Fc maps are colored green and red and contoured at +3 and -3 sigma respectively.

3) Given that the helix is proposed, and the tested TLK2 mutants were only somewhat informative (it was necessary to make a triple mutant in order to show a loss of the interaction), the validation (Figure 5) could have benefited from using a disruptive mutation on the interacting surface of TLK2. Further, the TLK2 peptide was not tested to determine whether it adopts alpha helical conformation in the presence and absence of Asf1. Moreover, do the TLK2 amino acid substitutions alter the helicity of the peptides? It is important to know that the loss of binding is due to lost interactions and not due to disruption of the peptides' structure. Circular dichroism would be able to demonstrate that 1) the TLK2 peptide exists as a helix in solution, 2) whether the helical structure is induced by binding to Asf1 in the complex and 3) the mutants did not perturb the peptide structure.

The reviewer raises several issues in this point. First, based on our use of a TLK2 L16A/R19A/F20A triple mutant they raise concerns that the TLK2 mutants were “only somewhat informative” and suggest that the validation “could have benefited from using a disruptive mutation on the interacting surface of TLK2”. In the original Fig 5 we showed that we also tested individual point mutations and that R11A and R12A each strongly inhibited binding. Nonetheless, we have now tested individual L16A, R19A and F20A mutations and see that single point mutations of L16A or R19A strongly inhibits ASF1 binding just like the triple mutant. The F20A also inhibits binding although it has a somewhat weaker effect than the other mutations. These data are now shown in **Supplemental Fig 5** and described on lines 279-281. We note that the results shown in Fig 5 also include individual ASF1 mutations at the interaction interface and that mutation of either ASF1 or TLK2 at the interface perturbs binding, thus we believe the mutagenesis strongly supports the mode of binding seen in the crystal structure.

The reviewer also suggests using circular dichroism to test whether the TLK2 peptide “adopts alpha helical conformation in the presence and absence of Asf1” and investigating whether the mutations used disrupt binding by altering helical propensity rather than binding. However, our crystal structure very strongly supports the idea that the peptide is helical when bound and mutagenesis of either the Asf1 or TLK2 supports the identification of the binding interface in the crystal structure. We see little reason to believe that each of the inhibitory single point mutations at the Asf1/TLK2 interface (R11A, R12A, L16A, R19A and F20A) act by disrupting helical propensity rather than directly perturbing Asf1 binding.

4) Given the interesting structure of Asf1 with the alpha helical TLK2 peptide (9-21), little description and few informative figures were provided. In fact, the artifactual site had similar attention. For example, the structural comparisons are limited to the Asf1 protein and no

superposition of the Asf1-H3 compared to the Asf21-TLK2 structure to illustrate the similarity of positions of the helices or key interacting residues was presented. It would be useful to determine how well conserved the positions are and whether there are any interactions that might be useful to discriminate between the binding interactions between Asf1 and of histones or TLK2. These might be the sites of informative mutants to address point 1 above.

We appreciate the comments from the referee suggesting additional presentation and analysis of the ASF1-TLK2 peptide complex, and for pointing out undue attention given to the artefactual binding site. We now include additional figure panels (**Supplementary Fig S4B** and **C**) showing a direct overly of the H3 region on the TLK2 peptide-ASF1 complex, which make clear the structural and sequence similarity of the corresponding regions. Similarity of the interacting residues on ASF1a is shown in the side-by-side comparison in **Fig 4**. We have also streamlined our discussion of the peptide modeled into the HIRA binding site to simply highlight why we suspected it to be an artefact. Finally, we have more extensively analyzed sequence conservation of TLKs at the ASF1 binding interface (new **Supplementary Fig S4A**). We found that four of five residues at the interface were similar in TLK orthologs from most animal and plant species, as well as a few protists where a TLK ortholog could be identified. The exception was Arg11, which is not conserved to histone H3. Consistent with this observation, the R11A mutant displayed a relatively modest (2-fold) decrease in binding to ASF1a compared to all other single mutants tested (**Fig 5A,B** and new **Fig S5**). Conservation of the ASF1-binding residues was particularly striking for invertebrates and plants where the overall conservation of the N-terminal region was low. We did find some groups whose TLK orthologs lacked this sequence – most notably insects in the order *Diptera* that includes fruit flies and mosquitos, though the sequence was conserved in all other insect groups we analyzed.

- 5) *The pHs and ionic strengths of the buffers used in different experiments is not the same and covers pH 7.5-8.0 and 100 mM to 250 mM salt, hampering direct or any quantitative comparisons.*

We agree that more uniform buffer conditions would perhaps allow more direct comparison between the various assays. We used conditions optimal in other systems for the specific assays performed; for example, there is reduced background signal in GFP nanobody pulldown experiments when performed at slightly acidic pH. We note that the DSF and pulldown assays are only semi-quantitative in nature and do not provide actual binding constants, and even the FP assays provide only apparent K_d values. What is clear from our data is that the trends we see are consistent across multiple types of assays performed under these different conditions, suggesting the general conclusions of the paper are robust.

- 6) Figures 1 use Asf1b, but the structural analyses were done with Asf1a. Is there a reason Asf1a wasn't used for all of the studies for consistency?

The goal of the Asf1 kinase assays in Figure 1 were to examine the impact of mutating residues flanking the phosphorylation site. We used Asf1b for these experiments because TLK2 phosphorylates it at only two sites, with the reported major site being a very close match to the consensus sequence as determined by peptide library analysis. Thus, we were able to establish the assay system by mutating a single minor site of phosphorylation; use of Asf1a would have required analysis of all possible triple mutants to determine the best background for our experiments. Ultimately given the purpose of the experiment, the choice of what specific phosphorylation site to use was arbitrary, and at the request of reviewer 2 we have now additionally included analysis of a synthetic consensus peptide that shares features with sites in both Asf1a and Asf1b. Finally, we note that the experiment in panel H comparing phosphorylation of full length Asf1b with the short C-terminal tail was also performed in with Asf1a (shown in **Fig 6**).

Minor:

1) There is no specific “histone chaperone domain” of Asf1 as mentioned lines 40, 50, 86-87, 159-160, 183, 186, 808 and 821 and Figure 1F. Asf1 is a histone chaperone and both the N-terminal globular core domain and the intrinsically disordered C-terminal tail have been shown to contribute to histone binding, which is the primary activity of a histone chaperone. Asf1-NT or globular domain or core domain would all be appropriate alternatives.

We thank the reviewer for pointing this out and we have corrected the text and figure legends to remove reference to “histone chaperone domain” – as suggested we instead refer to the N-terminal domain (Asf1-NT).

2) KD is not defined in Figure 1. Also, assuming it refers to kinase dead, why is the KD still active? Is Asf1a1-155 the same as Asf1a-NT? The amino acids in the constructs are not described in the materials and methods or the Figure legends, only in the text. This makes it difficult to interpret the figures. What is the numbering of the amino acids noted in the TLK2 LRF/AAA mutant. There is also mention of a TLK2-D592A mutant in Figure 1 - is this the KD? It would improve clarity if these could be defined either in the methods or the Figure legends where they are used in addition to the text.

We apologize for this oversight and for any confusion caused. We had used “KD” as an abbreviation for kinase dead. We have modified the text in the figure and legend to remove this abbreviation, and we now refer to the kinase inactive form only as TLK2-D592A. We have also provided the numbers of the TLK2 LRF/AAA mutant at various points in the text and legends and revised the text to ensure we use uniform terminology for ASF1.

The kinase inactive mutant preparation in **Fig 1** retains some activity – in this experiment the kinase expression construct (TLK2 Δ N178), which includes one of the proposed coiled-coil motifs, was purified from mammalian cells and we hypothesize that some associated endogenous kinase may account for the residual activity. Quantification of results as in **Fig 1D** shows that this is small compared to the activity of the non-mutated TLK2 Δ N178. In experiments with full-length TLK2 (**Fig 1F**), which contains three proposed coiled-coil motifs, the activity associated with the kinase dead is greater, consistent with ectopically expressed FL TLK2 co-purifying with higher levels of endogenous TLK2 as suggested by the reviewer. However, we note that the activity in our kinase inactive FL preparation is substantially less than for the wild-type construct.

3) Figures: The canonical coloration of the histone proteins is blue for H3 and green for H4. The coloration used here is different for the same proteins in different figures and even in different panels of the same figure. For example, In Fig 1 Asf1 is green. In Figure 4 Asf1 is grey, TLK2 is orange and H3 is cyan and H4 yellow. In Supplemental Fig 3, TLK2 is cyan (panel A and B), Asf1 is yellow green and orange, Asf1 is grey and TLK2 is orange (panel D). In Supplemental Fig 4, Asf1 is grey, TLK2 is pink and HIRA is green (panel A and B), in Panel C Asf1 is in grey and cyan and TLK2 is orange. This inconsistency makes it hard to discern the identities of the proteins by looking at the figure. Canonical colors ought to be used for the histones and consistent (different) colors be used for the other proteins.

We appreciate these comments and have revised our figures accordingly. ASF1a is now in grey in all figures except panel S4C in which the second neighboring ASF1a molecule is colored differently for contrast. The TLK2 peptide at the H3 binding site is beige, and purple at the HIRA site to distinguish between the two. H3/H4 are now in light blue/light green for consistency with canonical depictions.

4) Line 194 – *“The ASF1a structure appeared stable.” Crystallography cannot inform on protein stability.*

We have corrected our statement.

Reviewer #2

We greatly appreciate this reviewer’s support for publication of our manuscript. Three minor points were raised and we respond to each below and with modifications to the text of the manuscript.

Minor points:

1. *Fig. 2A, top panel: the D592A pulldown has two bands, the lower band is stronger, is it ASF1 degradation? Please explain.*

The multiple bands in the ASF1 blot represent different phospho-forms of the protein. Thus, expression of the active (but not the kinase inactive D592A) TLK2 shifts ASF1 towards the lower mobility phosphorylated form - this is now noted in the results section (Lines 159-161). This conclusion is further supported by new experiments in TLK2 knockout cells and in knockout cells reconstituted with wild-type or mutant TLK2 (**Fig 7**).

2. *Fig. 2B: Quantitation shows D592A interacted stronger, is the lower band counted as Asf1?*

Both ASF1 binds were counted in the quantification of binding in **Fig 2A**. We now clarify this in the figure legend.

3. *Methods, lines 376, 392 & 411: The E. coli expression strain should be BL21 (DE3) with RIPL or Rosetta, to reflect the incorporation of the T7 expression system.*

We have corrected the methods as suggested.

Reviewer #3

We thank this reviewer for their useful suggestions and constructive comments. As suggested, we have revised our manuscript to better reflect previous models and prior work and have added new experimental results in TLK2 knockout cells reconstituted with ASF1-binding defective mutants.

Specific comments

1. *the authors explore the TLK2 phosphorylation site motif and conclude that it has an unusual preference for acidic residues upstream of the phosphorylation site. The predictions from the*

motif analysis are tested on a single known phosphorylation site in Asf1b. This seems inadequate to make general conclusions and it is thus unclear whether the suggested motif is predictive for new TLK sites and substrates.

It would also be important to explore further the importance of the -1/-2 acidic residues on additional peptides. It is not entirely clear from the analysis that the acidic residues are more important than other residues. Would introduction of additional acidic residue on less strong sites improve phosphorylation?

In response to the reviewer's suggestion, we generated a consensus peptide substrate (NYDEETNHGWAKKK) that incorporates the most highly selected residues from the peptide array analysis at each position, including Glu residues at the -1 and -2 positions and Asn at the +1 position. We analyzed the relative rate of phosphorylation of this peptide in comparison with variants in which these key residues were individually substituted with alanine (data added as **Fig 1D**). We found that substitutions at the -2 and +1 positions reduced phosphorylation to a similar extent as Asf1b mutations to the corresponding positions. Substitution of the -1 Glu residue surprisingly trended toward increased phosphorylation but was not significantly different from the consensus peptide. These observations suggest that selectivity at the -1 position is dependent on the surrounding sequence context, but confirm that an acidic residue at the -2 position and a hydrophobic or amidic residue at the +1 position consistently promote phosphorylation in multiple sequence contexts.

As suggested by the reviewer, we also attempted to examine the impact of introducing an acidic residue into a non-consensus site. For these experiments, we examined phosphorylation by TLK2 of a peptide corresponding to Asf1a Ser175 (LQSLLSTDALPKKK), as well as two variants in which we introduced Glu residues at either the -2 or -1 position. However, we could not detect phosphorylation of any of these three peptides above background, even at comparatively high kinase concentrations and long incubation times. While this experiment clearly indicates that a single Glu residue is insufficient to promote phosphorylation by TLK2, we are unable to assess quantitatively the impact of such a residue in this context. These experiments are not included in the revised manuscript.

- 2. In the second part the authors identify an interaction between TLK2 and the Asf1b globular domain by apparent serendipity. However, this interaction was described many years ago by Klimovskaia et al., 2014 and led to the model that TLKs targets histone-free ASF1 for phosphorylation. The current work supports this model and provides the underlying structural basis for recognition of histone-free ASF1 by TLKs. This is interesting and important. However, the authors should demonstrate that this mechanism of H3 mimicry is important for TLK function and ASF1 phosphorylation in cells, e.g. by introducing truncated TLK2 lacking aa 2-23 into cells preferentially by genome editing or in a complementation system.*

We appreciate the reviewer's comments on the importance and interest of our study. In response to the reviewer's general remarks and to this specific point we have revised our manuscript to introduce the work of Klimovskaia et al., 2014 earlier in the manuscript. Furthermore, as suggested, we have assessed the consequences of CRISPR/Cas9-mediated loss of TLK2 and reconstitution with wild-type or mutant TLK2 on ASF1 phosphorylation and on clonogenic cell growth. Consistent with prior reports¹ we see that loss of TLK2 impairs both Asf1 phosphorylation (assessed by gel shift on SDS-PAGE) and cell growth and we show that reconstitution with wild-type TLK2 at least partially restores Asf1 phosphorylation and growth, but reconstitution with the kinase inactive D592A mutant, or with an Asf1-binding defective L16A/R19A/F20A TLK2 mutant impairs this rescue. Thus, the intact Asf1-binding site in TLK2 is important

for TLK2's ability to phosphorylate ASF1 in cells and for cell growth. These data are included in a new **Fig 7** and discussed on lines 316-339

3. *The authors do not discuss previous work showing that the yeast kinase Rad53 binds to several sites on scASF1 including the histone H3 binding surface and the B-domain binding sites (Jiao et al., 2012). Given the substantial similarity with the findings in the current work, it is unclear why this literature is left out.*

We thank the reviewer for raising this point – we now include discussion of similarities between TLK2-Asf1 interactions and the yeast Rad53-scASF1 interaction in the Discussion section of the manuscript.

Minor points:

1. *Figure 1F. There is residual activity in the TLK2 KD samples. Is the kinase purified from cells? in which case there is probably endogenous wt kinase present due to oligomerization of TLK1/2 endogenous and exogenous.*

The full-length kinase used in the original Fig 1F (now **Fig 1H**) was purified from mammalian cells. As TLK2 is known to oligomerize, we believe that as the reviewer suggests, associated endogenous TLK1/2 likely accounts for the residual activity. We now make note of this in the text (lines 146-149).

2. *Figure 2. Confirm that the peptide 2-23 is responsible for the major binding activity by comparing 34-772 with 1-772.*

As suggested, we generated an additional construct spanning TLK2 residues 24-772 and show in the revised **Fig 3** that this reduces ASF1 binding to background levels comparable to GFP or GFP TLK2 123-772

3. *Figure 3. Test whether H3.1 and B-domain peptides can act competitors. The prediction is that only the H3.1 peptide should work.*

As requested, we have now compared the abilities of H3.1 and B-domain peptides to compete with the TLK2 peptide for binding to ASF1-NT in the fluorescence polarization assay. These data (shown in new **Fig 5E**) confirm that the H3.1 peptide is an effective competitor while the B-domain peptide, like the non-binding TLK2 L16A/R19A/F20A triple mutant, does not compete with the labeled TLK2 peptide.

References

- 1 Kim, J. A. *et al.* Comprehensive functional analysis of the tousled-like kinase 2 frequently amplified in aggressive luminal breast cancers. *Nat Commun* **7**, 12991, doi:10.1038/ncomms12991 (2016).
- 2 Klimovskaia, I. M. *et al.* Tousled-like kinases phosphorylate Asf1 to promote histone supply during DNA replication. *Nat Commun* **5**, 3394, doi:10.1038/ncomms4394 (2014).
- 3 Segura-Bayona, S. *et al.* Differential requirements for Tousled-like kinases 1 and 2 in mammalian development. *Cell Death Differ* **24**, 1872-1885, doi:10.1038/cdd.2017.108 (2017).

REVIEWERS' COMMENTS

Reviewer #1 (Remarks to the Author):

The authors had adequately addressed my critique and I have no further comments.

Reviewer #3 (Remarks to the Author):

The authors have addressed all my concerns and I recommend the manuscript for publication after correcting a few minor points.

Minor points

- H3-H4 is common terminology for the dimer, please use this instead of the dot.
- The following statement is not correct (ASF1a/b depletion does not lead to fork collapse and DNA damage, the difference between ASF1a/b and TLK1/2 depletion in this respect is clearly stated in ref 39): TLK2 depletion causes replication fork collapse and consequent DNA damage, effects that are recapitulated by combined knockdown of ASF1a and ASF1b39.
- Please indicate the resolution of the crystal structure

Response to reviewer comments

We appreciate that the reviewers felt that we had addressed all their concerns in our revised manuscript.

Reviewer #3 raised 3 minor points which we have addressed in the newly revised manuscript.

- 1) - H3-H4 is common terminology for the dimer, please use this instead of the dot.

We now use the requested terminology throughout the re-revised manuscript

- 2) - The following statement is not correct (ASF1a/b depletion does not lead to fork collapse and DNA damage, the difference between ASF1a/b and TLK1/2 depletion in this respect is clearly stated in ref 39): TLK2 depletion causes replication fork collapse and consequent DNA damage, effects that are recapitulated by combined knockdown of ASF1a and ASF1b39.

We apologize for this error and have modified the sentence to correct our manuscript.

- 3) - Please indicate the resolution of the crystal structure.

We apologize for omitting this information and now state that the resolution for the apo-ASF1 structure is 2.1Å, and for the ASF1-TLK2 complex it is 2.71Å in table 1 and in the text.